# Possibility Before Utility: Learning And Using Hierarchical Affordances

**Robby Costales**[1]* **Shariq Iqbal**[1] **Fei Sha**[2]
[1]University of Southern California  [2]Google Research

## Abstract

Reinforcement learning algorithms struggle on tasks with complex hierarchical dependency structures. Humans and other intelligent agents do not waste time assessing the utility of every high-level action in existence, but instead only consider ones they deem *possible* in the first place. By focusing only on what is feasible, or "afforded", at the present moment, an agent can spend more time both evaluating the utility of and acting on what matters. To this end, we present *Hierarchical Affordance Learning* (HAL), a method that learns a model of *hierarchical affordances* in order to prune impossible subtasks for more effective learning. Existing works in hierarchical reinforcement learning provide agents with structural representations of subtasks but are not affordance-aware, and by grounding our definition of hierarchical affordances in the present *state*, our approach is more flexible than the multitude of approaches that ground their subtask dependencies in a *symbolic history*. While these logic-based methods often require complete knowledge of the subtask hierarchy, our approach is able to utilize incomplete and varying symbolic specifications. Furthermore, we demonstrate that relative to non-affordance-aware methods, HAL agents are better able to efficiently learn complex tasks, navigate environment stochasticity, and acquire diverse skills in the absence of extrinsic supervision—all of which are hallmarks of human learning.[1]

## 1 Introduction

Reinforcement learning (RL) methods have recently achieved success in a variety of historically difficult domains (Mnih et al., 2015; Silver et al., 2016; Vinyals et al., 2019), but they continue to struggle on complex hierarchical tasks. Human-like intelligent agents are able to succeed in such tasks through an innate understanding of what their environment enables them to do. In other words, they do not waste time attempting the impossible. Gibson (1977) coins the term "affordances" to articulate the observation that humans and other animals largely interpret the world around them in terms of which behaviors the environment *affords* them. While some previous works apply the concept of affordances to the RL setting, none of these methods easily translate to environments with hierarchical tasks. In this work, we introduce *Hierarchical Affordance Learning* (HAL), a method that addresses the challenges inherent to learning affordances over high-level subtasks, enabling more efficient learning in environments with complex subtask dependency structures.

Many real-world environments have an underlying *hierarchical* dependency structure (Fig. 1a), and successful completion of tasks in these environments requires understanding how to complete individual *subtasks* and knowing the relationships between them. Consider the task of preparing a simple pasta dish. Some sets of subtasks, like chopping vegetables or filling a pot with water, can be successfully performed in any order. However, there are many cases in which the dependencies between subtasks must be obeyed. For instance, it is inadvisable chop vegetables *after* having mixed them with the sauce, or to boil a pot of water *before* the pot is filled with water in the first place. Equipped with structural inductive biases that naturally allow for temporally extended reasoning over subtasks, hierarchical reinforcement learning (HRL) methods are well-suited for tasks with complex high-level dependencies.

---

*Correspondence to rscostal@usc.edu
[1]Code and videos of agent trajectories are available at https://github.com/robbycostales/HAL

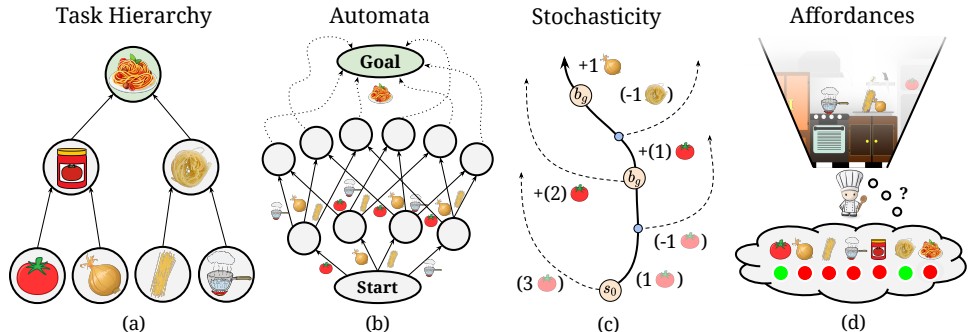

Figure 1: Many real world tasks, like making PASTA, can be conceptualized as a hierarchy (a) of subtasks. Automata-based approaches (b) map a history of subtask completion symbols to a context that indicates progress in the hierarchy. Approaches that assume symbolic history deterministically defines progress are not robust to stochastic changes in context (c) not provided symbolically. Hierarchical affordances (d) enable us to use incomplete symbolic information in the face of stochasticity by grounding context in the present state.

Existing HRL methods fall along a spectrum ranging from *flexible* approaches that discover useful subtasks automatically, to the *structured* approaches that provide some prior information about subtasks and their interdependencies. The former set of approaches (e.g. Vezhnevets et al., 2017; Eysenbach et al., 2018) have seen limited success, as the automatic identification of hierarchical abstractions is an open problem in deep learning (Hinton, 2021). But approaches that endow the agent with *more* structure, to make complex tasks feasible, do so at the cost of rigid assumptions. Methods that use finite automatas (Fig. 1b) to express subtask dependencies (e.g. Icarte et al., 2020) require the set of symbols, or *atomic propositions*, provided to the agent to be complete, in that the history of symbols maps deterministically to the current *context* (i.e. how much progress has been made; which subtasks are available). Importantly, these methods and many others (e.g. Andreas et al., 2017; Sohn et al., 2020) consider subtasks to be dependent merely on the completion of others.

Unfortunately, these assumptions do not hold in the real world (Fig. 1c). For instance, if one completes the subtask `cook noodles`, but they clumsily spill them all over the floor, are they now ready for the next subtask, `mix noodles and sauce`? While the subtask `cook noodles` is somehow *necessary* for this further subtask, it is not *sufficient* to have completed it in the past. The only way for automata-based approaches to handle this complexity is to introduce a new symbol that indicates that the subtask has been undone. This is possible, but extraordinarily restrictive, since, unless the set of symbols is complete, *none* of the subtask completion information can be used to reliably learn and utilize subtask dependencies. Modeling probabilistic transitions allows the symbolic *signal* to be incomplete, but still requires a complete *set* of symbols, in addition to predefined contexts. In order to make use of incomplete symbolic information, our approach instead learns a representation of context grounded in the present *state* to determine which subtasks are possible (Fig. 1d), rather than solely relying on *symbols*.

The contributions of this paper are as follows. First we introduce *milestones* (§4), which serve the dual purpose of *subgoals* for training *options* (Sutton et al., 1999) and as high-level *intents* (Kulkarni et al., 2016) for training our affordance model. Milestones are a flexible alternative to atomic propositions used in automata-based approaches, and they are easier to specify due to less rigid assumptions. Unlike a dense reward function, the milestone signal does not need to be scaled or balanced carefully to account for competing extrinsic motives. Next, we introduce *hierarchical affordances*, which can be defined over any arbitrary set of milestones, and describe HAL (§6), a method which learns and utilizes a model of hierarchical affordances to prune impossible subtasks. Finally, we demonstrate HAL's superior performance on two complex hierarchical tasks in terms of learning speed, robustness, generalizability, and ability to explore complex subtask hierarchies without extrinsic supervision, relative to baselines provided with the same information (§7.3).

## 2 RELATED WORK

Multi-task RL methods take advantage of shared task structure in order to generalize to new tasks from the same distribution (Andreas et al., 2017; Shiarlis et al., 2018; Devin et al., 2019; Sohn et al., 2020; Lu et al., 2021). Sohn et al. (2020) learn subtask preconditions, but use symbol-based contexts and do not learn and use their model of preconditions concurrently. Instead they assume a naive policy can sufficiently reach all subtasks. Furthermore, they assume ground-truth affordances

are provided at each step. Some works provide the agent with high-level task *sketches* (Andreas et al., 2017; Shiarlis et al., 2018) describing the order in which subtasks must be completed. While these sketches are advertised as "ungrounded" Andreas et al. (2017), they are in fact grounded by the inclusion of short sketches, which are the first to be introduced to the agent in a curriculum learning scheme (Bengio et al., 2009). Our approach instead uses a direct signal, which alone need not determine task progress, and can learn without exposure to other tasks with shared structure.

In using a set of discrete symbols to indicate subtask completion, our work is similar to the variety of approaches that apply *temporal logic* (TL) to the RL setting (Yuan et al., 2019; Hasanbeig et al., 2018; 2020; Li et al., 2017; 2018). These works typically provide the agent with a TL *formula*, as well as assignments of *atomic propositions* at each time-step. Some works use reward shaping to encourage satisfaction of the formula (Li et al., 2017; 2018), whereas others convert the TL formula to some finite state machine, which provides the agent with a structure that roughly expresses subtask dependencies (Hasanbeig et al., 2018; 2020; Yuan et al., 2019). Icarte et al. (2020) bypass this formula-to-automata conversion, and instead directly provide the automata to the agent in the form of a *reward machine* (RM). While RMs are more expressive than LTL formulas, they are less flexible than HAL, which can deal with incomplete sets of symbols, as well as context stochasticity.

Gibson (1977) introduces a theory of *affordances*, defined roughly as properties of the environment which must be measured relative to the agent. Heft (1989) and Chemero (2003) clarify affordances as *relations* between the agent and its environment. Khetarpal et al. (2020) formalize this relational definition of affordances in the context of RL, and model which low-level actions, given corresponding *intents*, are afforded in each state. In this work, milestones represent *high-level* intents corresponding to each subtask. They also demonstrate that modeling affordances speeds up and improves planning through the pruning of irrelevant actions, and allows for the learning of more accurate and generalizable partial world models. This approach does not directly translate to the hierarchical setting because subtasks, unlike actions, may fail for reasons other than affordances, meaning we do not have access to ground-truth affordance labels with which to train our model. Manoury et al. (2019) and Khazatsky et al. (2021) present approaches that can discover and use affordances to learn new skills, but their definition of affordances (i.e. a behavior is either afforded or not, with no notion of preconditions) does not translate to the hierarchical setting.

## 3 Preliminaries

Our setting involves learning behavior policies in *Markov Decision Processes* (MDP) using RL. An MDP is defined by the state space $\mathcal{S}$, action space $\mathcal{A}$, reward function $R : \mathcal{S} \times \mathcal{A} \to \mathbb{R}$, and state transition distribution $P : \mathcal{S} \times \mathcal{A} \to \triangle(\mathcal{S})$. The objective is to learn a policy, $\pi : \mathcal{S} \to \triangle(\mathcal{A})$, which selects actions that maximize expected future returns: $G(\pi) = \mathbb{E}\left[\sum_{t=0}^{\infty} \gamma^t r_t \,|\, a_t \sim \pi, s_t \sim P\right]$, where $\gamma$ is the discount factor. Sutton et al. (1999) introduce the *options framework*, which flexibly models hierarchical abstractions with minimal modification to the RL paradigm. Each *option*, $o := \langle \mathcal{I}_o, \pi_o, \beta_o \rangle$, is defined by an initiation set, $\mathcal{I}_o \subseteq \mathcal{S}$, indicating where the option can be selected, the corresponding option policy, $\pi_o$, and the termination condition, $\beta_o : \mathcal{S}^+ \to [0, 1]$, indicating the probability of termination in each state. Options turn our typical MDP into a *semi-Markov decision process (SMDP)* since the state transition distribution is, in general, no longer dependent on the current state and action, but also the present option, which was decided in a previous time-step. The design of this framework allows options to be treated similarly to actions, except they may be executed across multiple time-steps, interrupted, composed, and learned as separate subpolicies.

## 4 Milestones And Hierarchical Affordances

We consider tasks that can be decomposed into *subtasks*, each represented by a *milestone* symbol, $g \in \mathcal{G}$, where $\mathcal{G}$ is the set of symbols relevant to the task, and $|\mathcal{G}| = K$. For each subtask, we introduce a separate option, $\langle \mathcal{I}_g, \pi_g, \beta_g \rangle$, and we call $\pi_g$ a *subpolicy*. At each time-step, in addition to the extrinsic reward signal provided by the environment to indicate success on the overall task, we have access to a *milestone signal*, which is a vector $\boldsymbol{b}^t$ where each element $b_g^t \in \{0, 1\}$ indicates whether $g \in \mathcal{G}$ was completed on time-step $t$. In our PASTA example, we might receive a milestone each time we cut a vegetable, make the sauce, cook the noodles, etc. Milestones serve two main purposes. Firstly, milestones function as option *subgoals* (Sutton et al., 1999) that are in this work used to train each subpolicy (discussed in Section 5). Secondly, each milestone represents the *intent* of its corresponding subtask—similar to the action intents introduced to learn action-level affordances in the work of Khetarpal et al. (2020)—which we use to learn *hierarchical* affordances (discussed in

Section 6). In contrast to the standard options framework, primitive actions can only be executed as part of an option's subpolicy in our method. Generally, policies trained solely over options have no guarantee of optimality (Sutton et al., 1999), but we ensure the existence of an optimal solution by requiring $g_K \in \mathcal{G}$, where $g_K$ is the task's final milestone (indicating task success). When $\mathcal{G} = \{g_k\}$, our setting is standard, flat RL. Each additional milestone $g'$ added to $\mathcal{G}$ is useful as an intermediate signal so long as $g'$ corresponds to a unique behavior necessary for achieving $g_K$.

*Hierarchical affordances* are defined over $\mathcal{G}$ in the following way. The vector $\boldsymbol{f}^s = f^*(s)$ of size $K$ represents which milestones are *immediately* achievable from the present state $s$, without requiring the collection of any intermediate milestones, where $f$ is a *hierarchical affordance classifier*, and $f^*$ is the optimal one[2]. Formally, $f_g^s = 1$ if at time $t_0$ it is possible for future $b_g^T = 1$ without any $b_j^t = 1, j \neq g$, where $t_0 < t < T$. In PASTA, the milestone `mix cooked noodles and sauce` is not afforded at the beginning since `cook noodles` is required first. A successful policy trained within the vanilla options framework will *eventually* learn to execute options in contexts where they are most useful, regardless of each option's predefined initiation set. However, hierarchical affordances give us a principled way to directly adjust this set: for subtask $g$, we can set $\mathcal{I}_g = \{s \mid s \in \mathcal{S}, \ f_g^s = 1\}$. One can think of hierarchical affordances as using milestones to impose a state-grounded subtask dependency structure on top of the options framework, which we can use to prune impossible subtasks. If $\mathcal{G} = \{g', g_K\}$, an affordance-aware agent with access to optimal $f^*(s)$ will never initiate subtask $g_K$ from the beginning if $g'$ is a necessary intermediate behavior.

Some logic-based RL approaches (e.g. Yuan et al., 2019; Icarte et al., 2020) use *atomic propositions* as markers of subtask achievement to transition between contexts in a finite state machine. These approaches, and many other HRL works (e.g. Andreas et al., 2017; Sohn et al., 2020) define subtask preconditions in terms of *other subtasks*. There are two forms of stochasticity that hierarchical affordances, by virtue of being grounded in the present state, can more naturally address than symbolically-defined dependencies. We can conceptualize potential agent trajectories as graphs where *nodes* represent the attainment of milestones, and *edges* are the segments between them. *Node stochasticity* is affordance-affecting randomness that occurs either when milestones are attained (e.g. receiving varying quantity of an item), or at the beginning of the episode (i.e. starting in different contexts). *Edge stochasticity* is when affordances change at *any time* within a segment. We treat edge stochasticity events as infrequent *exceptions* to the typical subtask dependency rules. For example, after `cook noodles` is complete, `mix cooked noodles and sauce` is afforded, even if the agent may eventually spill the noodles on the floor. By grounding these rules in the current state, an affordance-aware agent can detect and adapt to edge anomalies. In Section 6, we describe in detail how hierarchical affordances are learned and used in stochastic environments where symbols alone would fail to reliably determine the current context.

## 5 LEARNING CONTROLLERS

Like h-DQN (Kulkarni et al., 2016), we use a *meta-controller* that selects the current subtask to attempt and a low-level *controller* which executes the subpolicy relevant to that subtask. The controller, $\pi : \mathcal{S} \times \mathcal{G} \to \triangle(\mathcal{A})$, selects low-level actions, $a \in \mathcal{A}$, given a state, $s \in \mathcal{S}$, and milestone, $g \in \mathcal{G}$, and aims to maximize the expected milestone signal rewards, $b_g$. Q-Learning (Watkins, 1989) trains these controllers by learning an estimate of the optimal Q-function: $Q_c^*(s, a; g) = \max_\pi \mathbb{E} \left[ \sum_{t=0}^{\infty} \gamma^t b_g^t \mid s_0 = s, a_0 = a, a_t \sim \pi, s_t \sim P \right]$ and deriving a policy from the Q-function as such: $\pi_g(a|s, g) = \mathbb{1}(a = \arg\max_{a'} Q_c(s, a'; g))$. Deep Q-Learning (Mnih et al., 2015) estimates $Q^*$ using deep neural networks. This Q-function is parameterized by $\theta = \{\theta_{\text{base}}, ..., \theta_g, ...\}$, where $\theta_{\text{base}}$ is a set of shared base parameters and $\theta_g$ is a goal-specific head. It is updated via gradient descent on the following loss function, derived from the original Q-learning update:

$$\mathcal{L}_{Q_c} = \mathbb{E}_{(s_t, a_t, r_t, s_{t+1}, g_t) \sim D_c} \left[ \left( Q_c(s_t, a_t; g_t, \theta) - b_g^t - \max_{a_{t+1}} Q_c(s_{t+1}, a_{t+1}; g_t, \bar{\theta}) \right)^2 \right] \quad (1)$$

where $D_c$ is a replay buffer that stores previously collected transitions, and $\bar{\theta}$ are the parameters of a periodically updated target network. Both of these components are included to avoid the instability associated with using function approximation in Q-Learning.

The meta-controller $\Pi : \mathcal{S} \to \triangle(\mathcal{G})$ aims to execute subtasks to maximize extrinsic rewards received by the environment. Again, we estimate a Q-function, this time over a dilated time scale

---

[2]Unlike option *completion* predictions (Precup et al., 1998), affordances predict possibility of *success*.

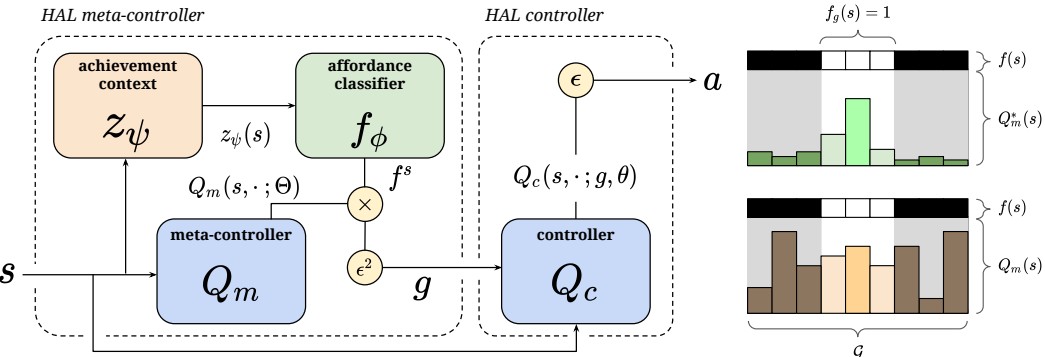

Figure 2: Left: Architecture diagram for complete HAL method. Q-values of the meta-controller are masked by the output of the affordance classifier. The $\epsilon$ operator represents the standard $\epsilon$-greedy action selection procedure used in Q-learning, while $\epsilon^2$ represents our affordance aware version. Right: For an optimal policy (top), the mask will have no effect since Q values will naturally be low for unafforded subtasks. However, a suboptimal policy (bottom) will benefit from a mask since it can be efficiently learned and used to prune irrelevant subtasks before TD errors can propagate.

(i.e. we allow the low-level controllers to run for multiple steps before choosing new goals): $Q_{\text{mc}}^*(s, g) = \mathbb{E}\left[\sum_{t=0}^{N} r_t + \max_{g'} Q_{\text{mc}}^*(s_N, g') \mid s_0 = s, g_0 = g, a_t \sim \pi_g, s_t \sim P\right]$, where $N$ is the (variable) number of steps the option runs for. When collecting data in the environment, we add transitions $(s_t, g_t, \sum_t^{t+N} r_t, s_{t+N})$ to a separate meta-replay buffer, $D_{\text{mc}}$, used to train our meta-Q-function (parameterized by $\Theta$) with a loss similar to Eq. 1, but without any goal-conditioning. In Section 6 we describe how hierarchical affordances are integrated into this training procedure.

## 6 HIERARCHICAL AFFORDANCE LEARNING

In typical HRL methods, if the meta-controller is yet to receive extrinsic reward from the environment, there will be no preference for selecting any subtask over the others. However, by restricting the selection of subtasks to ones that have proven merely to be *possible*, an agent can avoid wasting time attempting the impossible, and reach more fruitful subtasks faster. Suppose from experience, gained through random exploration, the agent achieves milestone $g$ (where achievement means $b_g = 1$) very often from the initial state set $\mathcal{I}$, but never $j \in \mathcal{G}$, despite being able to achieve $j$ in later states. With enough experience, the agent should become confident that $j$ is not achievable without completing other milestones first, and should not bother selecting $j$ from any $s \in \mathcal{I}$, while $g$, and any others that *are* achievable from those states, should instead be considered. If we had access to an oracle function, $f^*(s)$, that accurately computes hierarchical affordances for our task, we could prune impossible subtasks by masking the otherwise uninformed policy with the affordance oracle output: $p(g|s) \propto f_g^s * \Pi_\Theta(g|s)$. In the following sections, we describe a method that can learn an approximate $f(s) \approx f^*(s)$ from experience and leverage it in real-time for more effective learning.

**The false negative problem** Recall that $f(s)$ outputs a vector $\boldsymbol{f}^s$, where each $\boldsymbol{f}_g^s$ indicates the possibility of collecting milestone $g$ from state $s$ without requiring intermediate milestones. To train each binary classification head, $f_g(s)$, we must somehow generate labeled data for each milestone. Suppose an option was initialized at time $t_o$, and in time $T$ milestone $g$ is received. If no others were received since the start of the option, we may assume that for any $t$ where $t_o \leq t \leq T$, the collection of milestone $g$ was afforded, so we can use the set of states $\{s_{t_o}, s_{t_o+1}, \ldots, s_T\}$ as positive (i.e. $f_g^*(s) = 1$) training examples for the affordance classifier. Even if $g$ was not the intended milestone, we can still generate positive training examples for $f_g$ in this way. If an option has failed to collect intended milestone $g$, either through timing out or collecting an unintended milestone, we might be tempted to use the states encountered during that option as negative examples (i.e. $f_g^*(s) = 0$). However, this occurrence can either be indicative of the states not affording $g$, or that the subpolicy corresponding to $g$ is sub-optimal and has failed despite $g$ being afforded. These false negatives are a problem for any approach requiring function approximation via neural networks, which are generally not robust to label noise (Song et al., 2020). It is particularly troublesome in our case since the noise is greater than the true signal when the subpolicies are under-trained.

**Context learning**  Suppose we had access to an abstract state representation $\boldsymbol{z}_{\mathrm{aff}}^s = z_{\mathrm{aff}}^*(s)$, where any states $s_i$ and $s_j$ are mapped to the same value only when $f^*(s_i) = f^*(s_j)$. With a representation that could cluster states in this way, we could trivially determine the falsity of a collected negative $s \in \mathcal{D}_g^-$ by checking if $z_{\mathrm{aff}}^*(s) = z_{\mathrm{aff}}^*(s_j)$ for any $s_j \in \mathcal{D}_g^+$, that is, if we have encountered a true positive with the same representation. The classification procedure could be interpreted as "labeling" these contexts with affordance values. This is somewhat of a "chicken and egg" problem, since to learn affordances, we require a representation that maps states to contexts with the same affordance values, which clearly requires some prior knowledge about affordances. Fortunately, from Section 4, we know that affordances will only change when either (1) a milestone is collected and (2) when edge stochasticity occurs. Since (2) is by definition a rare occurrence, states $s_t$ and $s_{t+1}$ are more likely than not to satisfy $f^*(s_t) = f^*(s_{t+1})$, so long as they exist in the same *segment* between milestones. In this case, we can say that $s_t$ and $s_{t+1}$ share the same *achievement context*, $\boldsymbol{z}^s = z(s)$. Let $z_\psi(s)$ be an achievement context embedding represented by a differentiable function parameterized by $\psi$. We can train $z_\psi(s)$ from experience using the following contrastive loss:

$$\mathcal{L}_\psi = \sum_j \left[ \left\| z_\psi(s_j^a) - z_\psi(s_j^p) \right\|_2^2 - \left\| z_\psi(s_j^a) - z_\psi(s_j^n) \right\|_2^2 + \alpha \right]_+ ,$$

where each $s_j^a$ is a randomly chosen *anchor*, each $s_j^p$ is a *positive*[3] example chosen within the segment according to a (truncated) normal distribution, $\mathcal{N}_T(0, \sigma^2)$, centered around (and excluding) $s_j^a$, $s_j^n$ is chosen randomly among other segments and is treated as a *negative* example, and $\alpha$ is an arbitrary margin value. This loss pushes representations of states from the same achievement context together, and pulls representations of states from different contexts apart. We show in Appendix B that a wide range of $\sigma$ produce useful representations. For our edge stochasticity experiments (Figure 5) we use a low $\sigma = 2.0$ to reduce the risk of sampling across affordance changes.

**False negative filtering**  In the learned representation space, we expect false negative points to be closer to positive points than true negatives. Given a negatively-labeled state $s_q$ for classifier head $f_g$, we compute[4] the mean distance from $z_\psi(s_q)$ to the representations of the $k$ closest positive points in a population uniformly sampled[5] from $\mathcal{D}_g^+$, denoted $d_q^k$. We expect $d_q^k$ to be large for true negatives and small for false ones, and we can determine an effective separating margin in the following way. First we compute distance scores for a random sample of positive points, denoted $\{d_p^k\}$, to use as reference. We ensure these points come from segments that are disjoint from the population points' segments to avoid trivially low scores that might skew the distribution. We then fit a Gaussian distribution to $\{d_p^k\}$ and compute an upper confidence bound $\rho$ for a given percentile value and confidence level. Any $d_q^k < \rho$ is very similar to positive points in the representation space, so we count $s_q$ as a false negative and exclude it from our training set. Note, we do not train head $f_g$ (and therefore do not reliably prune) until we have access to both positives and negatives for $g$.

**Method overview**  The HAL architecture consists of a bi-level policy like h-DQN, a context embedding network, and an affordance classifier (see Figure 2), which are all learned concurrently (full algorithm in Appendix E). Intuitively, the affordance classifier is able to generalize to a novel state, $s$, by first identifying the abstract achievement context, $z_\psi(s)$, associated with the state, and then outputting an affordance value based on previous experience in that context. If $z_\psi(s)$ has also not been encountered, that context will not be strongly "labeled" either way, so we will not be invariably pruning it. The meta-controller selects a subtask $g$ at the beginning of the episode, and selects a new subtask $g'$ after collecting *any* milestone or whenever an option times out after a predefined number of steps (our $\beta_g$). At each step, the current state is fed to the controller, which outputs an action conditioned on the most recently selected subtask. After discretizing the classifier's output to a binary mask, we perform an affordance aware version of $\epsilon$-greedy as follows. Given parameters $\epsilon_{\mathrm{aff}}$ and $\epsilon_{\mathrm{mc}}$, we select a random subtask within the mask with probability $\epsilon_{\mathrm{aff}}$, randomly across all subtasks with probability $\epsilon_{\mathrm{mc}}$, and otherwise select greedily with respect to meta-Q *within* the mask.

## 7  EXPERIMENTS

In our experiments we aim to answer the following questions: (1) Does HAL improve learning in tasks with complex dependencies? (2) Is HAL robust to milestone selection and context stochasticity? (3) Can HAL more effectively learn a diverse set of skills when trained task-agnostically?

---

[3]Here, the usage of "positive" and "negative" refers to whether points share the same achievement context.

[4]This procedure is akin to the particle entropy approach used in (Liu & Abbeel, 2021).

[5]For efficiency, we sample just enough points so that we are likely to cover all encountered contexts.

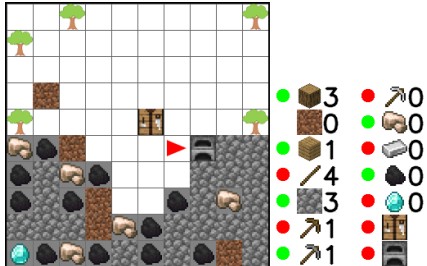 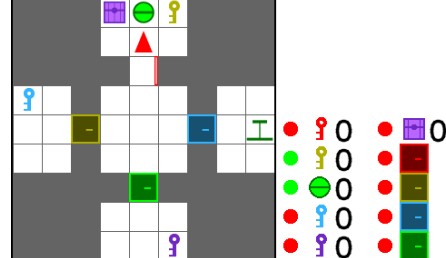

Figure 3: Screenshots of CRAFTING (left) and TREASURE (right) environments. Displayed to the right of the environments are each item's ground-truth affordance indicator and inventory count.

## 7.1 ENVIRONMENTS

We evaluate our method, along with several baselines, on two complex environments with intricate subtask dependency structures: CRAFTING and TREASURE. Both environments (visualized in Figure 3) are extensions of the minigrid framework (Chevalier-Boisvert et al., 2018). Agents receive an egocentric image of the environment, as well as a vector describing their inventory (items picked up from the environment and currently in their possession) as observations. The action spaces are discrete and include actions for turning left/right, and moving forward/backward, as well as environment specific actions detailed below. Task hierarchies, walk-throughs of successful trajectories, and additional information about both environments are included in Appendix A.

**CRAFTING** is based on Minecraft, a popular open-ended video game in which players collect resources from their environment and use them to craft objects which can be used to then obtain more resources. As such, the hierarchy of possible subtasks is immensely complex and presents a significant challenge for AI agents to reach subtasks deeper in the hierarchy. We develop an environment that replicates this hierarchical complexity without the commensurate visuomotor complexity which our method does not aim to address. In addition to movement actions, CRAFTING includes actions to mine the object immediately in front of the agent (which requires an appropriate pickaxe), as well as to craft and smelt the various objects (pickaxes, iron ingots, etc.). The full set of milestones contains items that are either craftable or collectable. CRAFTING naturally contains node stochasticity since the collection of certain items, due to the random procedural generation, requires slightly different milestone trajectories across episodes (e.g. mining variable amount of stone to encounter diamond).

**TREASURE** is a navigation task that requires the agent to collect various items and use them to unlock rooms to reach further items. The ultimate goal is to unlock a treasure chest, which requires a sequence of collecting several keys, as well as placing an object on a weight-based sensor, in order to open the requisite doors. Agents can only carry one object at a time, so they must reason about which object to pick up based on what it will afford them (e.g. if the weight-based sensor room is locked, the weight object is not currently useful). Like CRAFTING, TREASURE contains node stochasticity due to the procedural generation. For example, the central room that the agent is spawned in can contain either of the red or yellow key individually, or both together. Unlike CRAFTING, which has a large action space to accommodate the various crafting recipes, this environment only contains actions to move and a single "interaction" action that is used to pick up keys, open doors, etc. While CRAFTING has a more complex hierarchy and greater diversity in the potential ordering of subtasks, TREASURE has on average more difficult subtasks. The full set of milestones contains each object the agent can successfully interact with in the environment (e.g. opening door, collecting key).

## 7.2 BASELINES

Our set of baselines is summarized in Table 1. All methods are based on the Rainbow (Hessel et al., 2018) Deep Q-Learning algorithm, which combines several improvements to vanilla DQNs (Mnih et al., 2015). To compensate for the lack of milestone signals, non-hierarchical methods use a dense reward function that incorporates milestone signals for the first time each milestone is obtained in the episode. To evaluate the efficacy of our affordance classifier

Table 1: Summary of baselines.

|  | Hier-archical Agent | Afford-ance Mask | Hind-sight Replay | False Negative Filtering |
|---|---|---|---|---|
| Oracle | ✓ | *Truth* | ✓ | N/A |
| HAL (ours) | ✓ | Learned | ✓ | ✓ |
| HAL(–FNF) | ✓ | Learned | ✓ | ✗ |
| H-Rainbow | ✓ | N/A | ✗ | N/A |
| H-Rainbow (+HER) | ✓ | N/A | ✓ | N/A |
| Rainbow | ✗ | N/A | N/A | N/A |

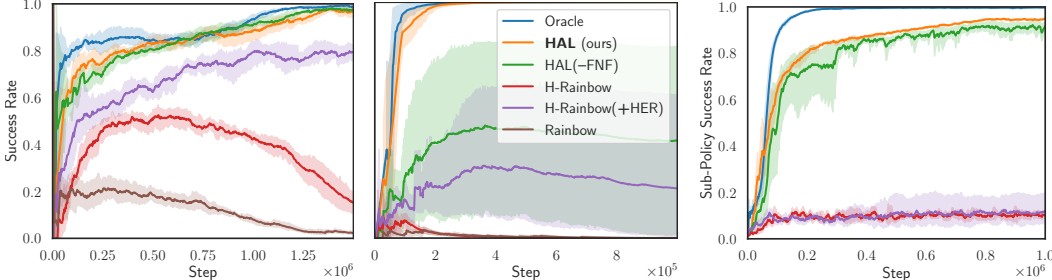

Figure 4: Success rate over the course of training for CRAFTING iron task (left) and TREASURE (center). Sub-policy success for TREASURE (right). Success rate is the proportion of episode where the agent receives the target milestone, and sub-policy success is how often sub-policies, on average, receive the correct milestone when called, before a timing out or collecting incorrect milestones.

online learning procedure, we compare our method to an "oracle" that differs only by using *ground-truth* affordances for masking subtasks. The node stochasticity inherent to both environments, as well as the edge stochasticity later explored, preclude the use of any methods that reason solely over symbols (i.e. automata-, sketch-, or subtask dependency-based approaches). We also incorporate a version of Hindsight Experience Replay (HER) (Andrychowicz et al., 2017) adapted for the discrete milestone setting, which involves re-using "failed" trajectories that result in the collection of an un-intended milestone (given the selected subpolicy) as successful data for the relevant subpolicy. For instance, if an agent accidentally collects iron while executing its wooden pickaxe sub-policy, it can use this trajectory to train its iron sub-policy.

## 7.3 RESULTS

**Learning efficacy** First, we evaluate the ability of HAL and our baselines to learn successful policies for complex tasks in each environment. Learning curves are shown in Figure 4 (plots depict mean and 95% confidence interval over 5 seeds). In both environments, HAL significantly outperforms the strongest baseline, H-Rainbow(+HER) (HR+H), despite both methods receiving the same information, and performs only slightly worse than the oracle, which has access to ground truth. Incorporating HER into H-Rainbow leads to a significant improvement. False negative filtering (and all other HAL components; see Appendix B) appears crucial for learning in the TREASURE environment, but not as much for CRAFTING, though in both cases filtering improves mask accuracy. Removing false negative filtering causes HAL(-FNF) to be pessimistic (i.e. over-pruning subtasks, see Figure 15 in Appendix D), ultimately leading to its unstable learning. Since masking impossible subpolicies would have no impact on an optimal meta-controller, HAL's success must stem from its ability to learn a useful mask *before* TD errors are able to propagate through the meta-controller's Q function. HAL utilizes a more easily learnable function (affordance classifier) to reduce the amount of unnecessary expensive learning (TD error propagation) required. We see from Figure 15 that, throughout training, HAL's mask has an impact on greedy subtask selection ∼60% of the time, which is evidence that HAL avoids wasting time learning Q-values that the mask is able to prune. Lastly, because affordance-aware methods are more likely to initiate subtasks in an appropriate context, we see they achieve a significantly higher average subpolicy success rate (Figure 4, right).

**Robustness to milestone selection** In this section we evaluate HAL's robustness to the selection of milestones. Affordances change when a milestone is removed since that milestone no longer acts as an intermediate link between others. One downside of some approaches that use symbol-based contexts is that an entirely different automata or subtask dependency graph must be defined over the new set of symbols. HAL does not use prior information of this kind, so the learning process is the same across all sets. Figure 5 shows HAL's success on the CRAFTING environment's iron task when using "incomplete" milestone sets, relative to the full human-designed set. We see that randomly removing 1 milestone makes no significant difference for HAL, and even after removing 4 milestones, HAL still achieves better performance than HR+H using the full set. HAL's performance drops when 5 milestones are removed likely due to the increased sparsity of the signal (i.e. greater subtask length) and variance in milestone set quality. However, when we double the training time for these same sets, we find that HAL is able to converge to a 97% success rate on at least one set, while HR+H fails to converge on any set and ends with a maximum success rate of around 70%.

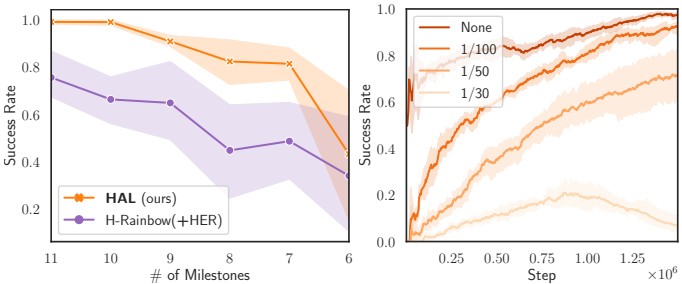
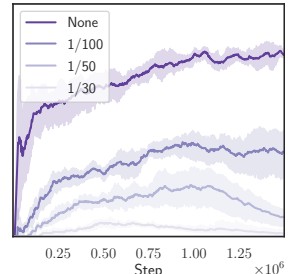

Figure 5: Comparing the robustness of HAL and HR+H to varying milestone sets (left) and various edge stochasticity frequencies (center and right, respectively) in CRAFTING iron task.

**Robustness to stochasticity**  We modify CRAFTING so that at each environment step, there is a certain probability that an item in the inventory will disappear. In order to make the task feasible, rare items are less likely to disappear than common ones. This procedure produces *edge stochasticity*, since the disappearance of items may alter affordances, and this can occur at any time. We test three different levels of stochasticity, and display the learning curves in Figure 5. With a disappearance frequency of $1/100$ that cuts HR+H's success rate in half, HAL is still able to reach its non-stochastic success rate. With a frequency of $1/50$, HAL performs comparably to HR+H with no stochasticity. To put these stochasticity rates into context, the algorithm's average episode length about halfway through training is still over 1000 steps (see Figure 14 in Appendix C), meaning *dozens* of items are removed from the agent's inventory over the course of an episode. By learning a model of affordances grounded in the present state, HAL is able to detect and adapt to these stochastic events.

**Task-agnostic learning**  We next test the ability of HAL to learn skills when no task-specific extrinsic rewards (only milestones) are provided by the environment. Since we cannot learn a meta-controller in the absence of rewards, we instead randomly select subtasks with some probability $\epsilon_{\mathrm{mc}}$, and random *afforded* subtasks otherwise (only for HAL). We evaluate both HAL and HR+H. In Figure 6 we see that by the end of $10^6$ steps, HAL is able to more reliably complete the milestones deeper in the hierarchy in the CRAFTING environment. We note that HR+H is able to marginally outperform HAL in tasks shallower in the hierarchy (e.g. wood pickaxe, stone, furnace), potentially as a result of failing to reach deeper tasks and getting more practice on shallower ones. This result is an indication of the general utility of HAL in environments with complex task hierarchies.

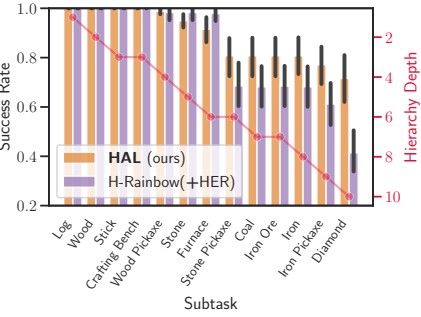

Figure 6: Percentage of episodes where each milestone is achieved in CRAFTING environment task-agnostic setting.

## 8   CONCLUSION AND FUTURE WORK

The present work can be viewed as a first step towards bridging the substantial gap between flexible hierarchical approaches that are currently intractable, and methods that impose useful structures but are too rigid to be of practical use. We introduce HAL, a method that is able to utilize incomplete symbolic information in order to learn a more general form of subtask dependency. By grounding subtask dependencies in the present state by learning a model of hierarchical affordances, HAL is able to navigate stochastic environments that approaches relying solely on symbolic history are unable to. We demonstrate that HAL learns more effectively than baselines provided with the same information, is more robust to milestone selection and affordance stochasticity, and can more thoroughly explore the environment's subtask hierarchy. Given HAL's flexible formulation and success in the face of incomplete and stochastic symbolic information, we foresee future work integrating HAL with option (or subgoal) discovery methods (e.g. Bacon et al., 2017; Machado et al., 2017; Bagaria & Konidaris, 2019) to obtain performance gains in complex tasks without requiring pre-specified milestones. Additionally, future work might be able to extend HAL to continuous goal-spaces, but this would require revising the definition of hierarchical affordances provided here, as it currently requires a notion of intermediate subgoal completion.

## 9 ACKNOWLEDGEMENTS

We thank Natasha Jaques, Sébastien Arnold, and the anonymous reviewers for their feedback on mature drafts of this manuscript. This work is partially supported by NSF Awards IIS-1513966/ 1632803/1833137, CCF-1139148, DARPA Award#: FA8750-18-2-0117, FA8750-19-1-0504, DARPAD3M - Award UCB-00009528, Google Research Awards, gifts from Facebook and Netflix, and ARO# W911NF-12- 1-0241 and W911NF-15-1-0484.

## 10 REPRODUCIBILITY STATEMENT

All code for environments, HAL, and other relevant baseline algorithms is provided at the following link: https://github.com/robbycostales/HAL. We provide instructions for installing the necessary dependencies, and enumerate commands that allow researchers to replicate results in this paper. Most relevant hyperparameters and additional implementation details are also listed in the Appendix.

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

# Appendix

## Table of Contents

## A  ADDITIONAL ENVIRONMENT DETAILS

In both environments, extrinsic rewards are a sparse binary signal provided upon the completion of a final goal. Milestone signals are similarly formulated for each of the possible subtasks. For both types of rewards (extrinsic and milestone) we introduce a per step penalty that encourages agents to achieve their goal as quickly as possible. TREASURE and CRAFTING environments are extensions of the minigrid framework (Chevalier-Boisvert et al., 2018).

### A.1  CRAFTING

As discussed in Section 7.3, our CRAFTING environment is designed to maximize hierarchical complexity while minimizing visuomotor complexity, which we do not aim to address with our method. Similar Minecraft-based environments have been used in the literature; however they do not meet these requirements. The MineRL environment (Guss et al., 2019) provides an interface into the full game of Minecraft; however, effective behavior in this environment requires learning complex visuomotor policies in addition to understanding the hierarchical relationships between subtasks. In order to evaluate our method effectively, we only aim to test the latter. Other work has used Minecraft-inspired environments (Andreas et al., 2016; Sohn et al., 2020); however, these versions involve simplified subtask hierarchies. Our environment replicates the complexity of the MineRL subtask hierarchy while remaining perceptually simple. A concurrently developed environment contains similar hierarchical complexity while minimizing perceptual complexity (Hafner, 2021).

Figure 7 displays an abstract representation of the CRAFTING environment subtask hierarchy. Each arrow points from one subtask to another, where the latter subtask requires the former in some way. The CRAFTING environment contains a variety of dependency types. Many items must be built or *crafted* from other resources. Some of these require the agent to be within the vicinity of a crafting bench. Other items require specific pickaxes to be mined, and yet others must be smelted in a furnace using a raw material and a fuel source. Another complexity missing from Figure 7 is that crafting *recipes* require specific amounts of items. For instance, in our environment a stone pickaxe requires three stone and two sticks. Each stone must be mined individually, resulting in three separate milestones, while just two wood are required to craft four sticks. Lastly, there are

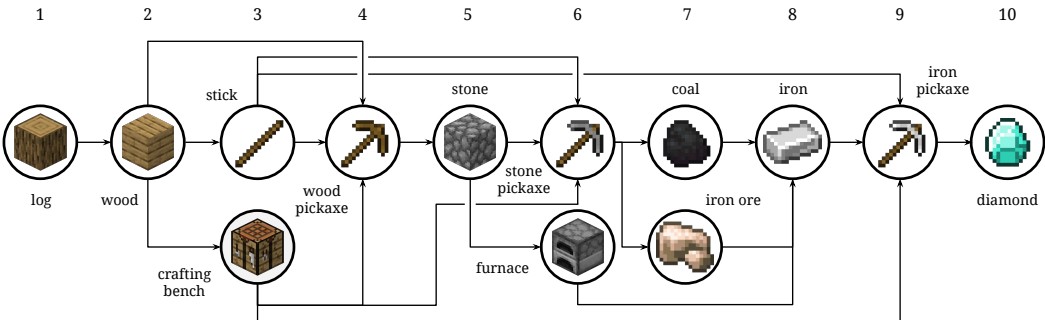

Figure 7: Abstract representation of CRAFTING subtask hierarchy, where milestones are circled and arrows indicate to explicit subtask dependencies between milestones and numbers indicate the depth of each item in the hierarchy (used in Figure 6). Numerical preconditions for crafting recipes are not shown, as well as other possible implicit environmental dependencies (e.g. mining $x$ to reach $y$).

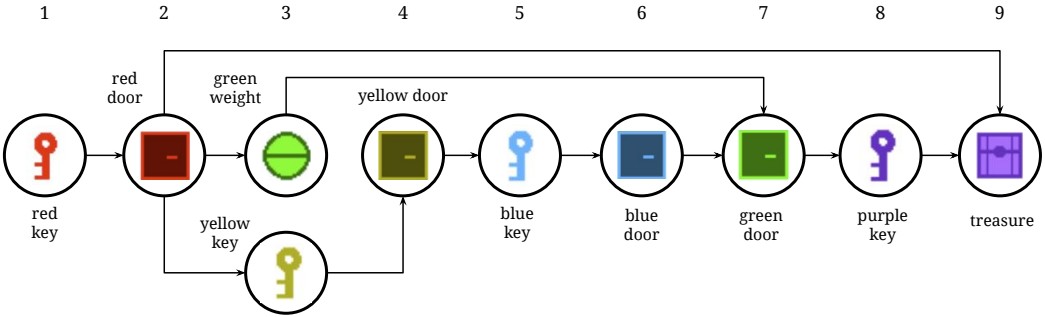

Figure 8: Abstract representation of *one* possible TREASURE subtask hierarchy, where milestones are circled and arrows indicate to explicit subtask dependencies between milestones.

other implicit dependencies than the ones shown, depending on the definition of the milestones set. For example, although the only official prerequisite for obtaining diamond is having an iron pickaxe, in practice, an agent will need to mine stone and other blocks to reach the diamond, and the successful mining of each of these blocks may be considered a milestone.

CRAFTING is procedurally generated in the following way. For the lower half of the environment, each cell is randomly assigned stone, coal, iron, or dirt to each cell with varying probabilities. Diamond is randomly assigned to a cell in the bottom-most layer. Trees (from which logs are obtained) are abundantly scattered in the upper half, and a few irrelevant dirt blocks are placed in this region as well. If the generated environment does not have enough resources, it is regenerated with the next random seed.

## A.2 TREASURE

In Figure 8 one *possible* TREASURE subtask hierarchy is displayed. Unlike CRAFTING, there are multiple different abstract hierarchies depending on the initial state of the environment. At the beginning of each episode, either both keys are available, only the yellow key, or only the red key (the instance shown in Figure 8). While the environment dynamics remain the same across all episodes, the agent must infer which hierarchy is appropriate based on the initial configuration of the environment. TREASURE is procedurally generated by randomly assigning each colored door to the room entrances, randomly determining which keys are accessible from the starting room, and lastly placing all objects behind their appropriate doors at random positions within the room.

## A.3 WALK-THROUGHS

Successful human walk-throughs for both TREASURE and CRAFTING environments are described in Figures 9 and 10 respectively.

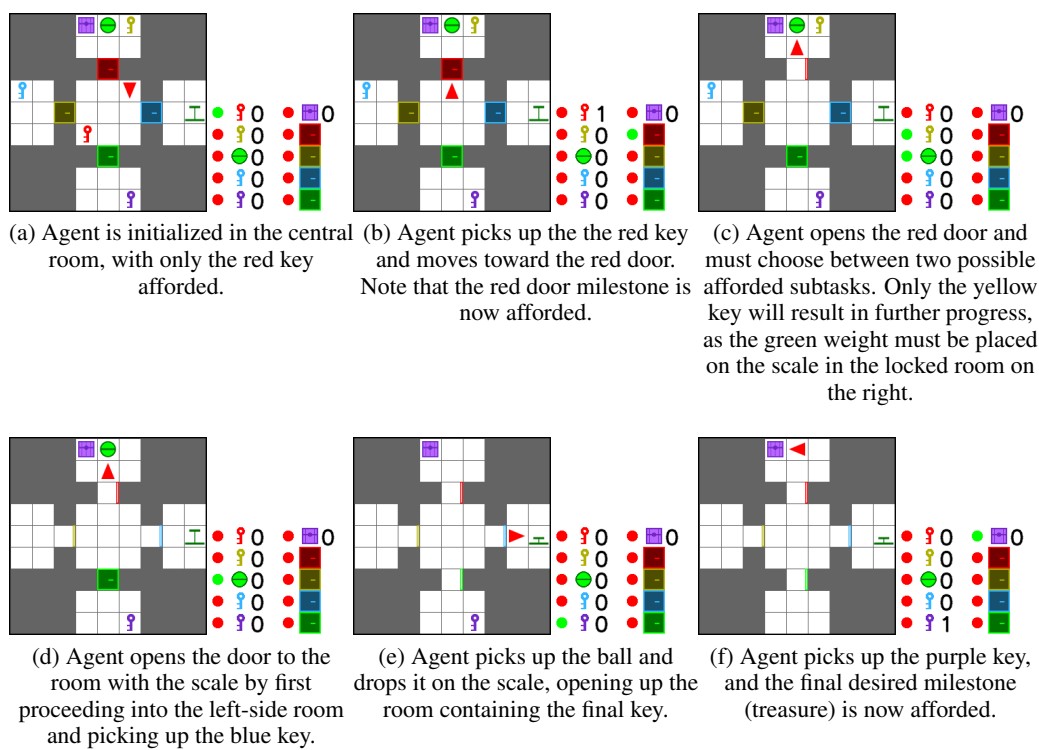

(a) Agent is initialized in the central room, with only the red key afforded.

(b) Agent picks up the the red key and moves toward the red door. Note that the red door milestone is now afforded.

(c) Agent opens the red door and must choose between two possible afforded subtasks. Only the yellow key will result in further progress, as the green weight must be placed on the scale in the locked room on the right.

(d) Agent opens the door to the room with the scale by first proceeding into the left-side room and picking up the blue key.

(e) Agent picks up the ball and drops it on the scale, opening up the room containing the final key.

(f) Agent picks up the purple key, and the final desired milestone (treasure) is now afforded.

Figure 9: Walk-through of a successful TREASURE task episode. Items currently in the agent's possession are indicated by the numbers on the right hand side and ground-truth affordances are indicated by green circles if the milestone is afforded and red if not.

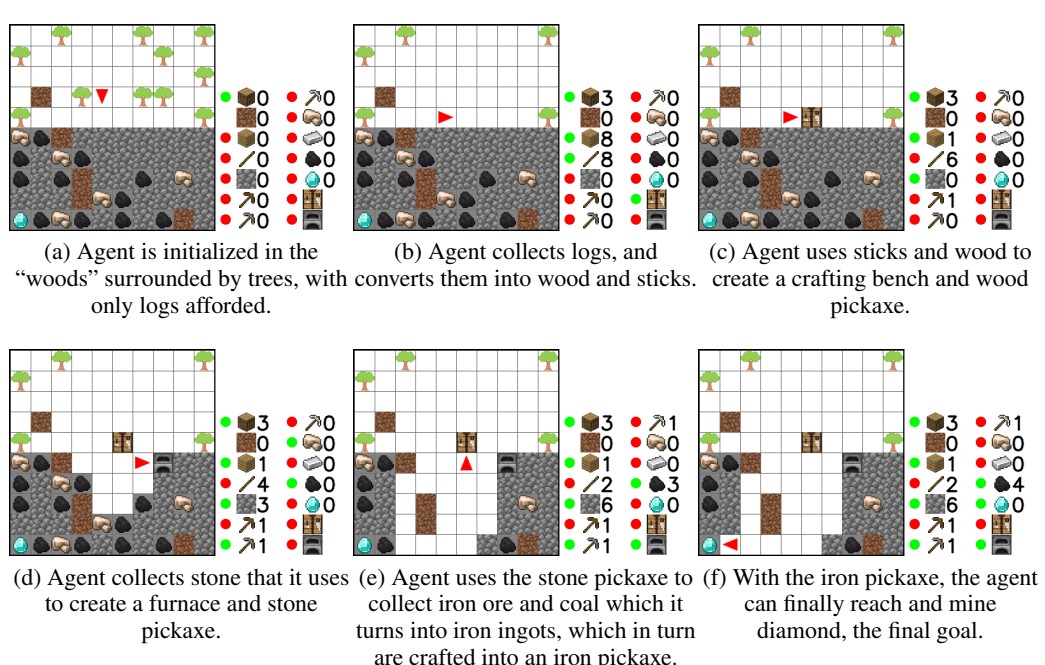

(a) Agent is initialized in the "woods" surrounded by trees, with only logs afforded.

(b) Agent collects logs, and converts them into wood and sticks.

(c) Agent uses sticks and wood to create a crafting bench and wood pickaxe.

(d) Agent collects stone that it uses to create a furnace and stone pickaxe.

(e) Agent uses the stone pickaxe to collect iron ore and coal which it turns into iron ingots, which in turn are crafted into an iron pickaxe.

(f) With the iron pickaxe, the agent can finally reach and mine diamond, the final goal.

Figure 10: Walk-through of successful CRAFTING environment diamond task episode.

# B ABLATION AND ROBUSTNESS RESULTS

## B.1 COMPONENT ABLATIONS

In Figure 11 we plot success on TREASURE after removing various integral components of the full HAL method. We find that all components introduced in this work are necessary for achieving the best performance on this task. The only variation that also reliably converges to 100% success rate is when the affordance classifier is provided with the raw state as input rather than using the learned representation, but this variation is undesirable since more learnable parameters are introduced.

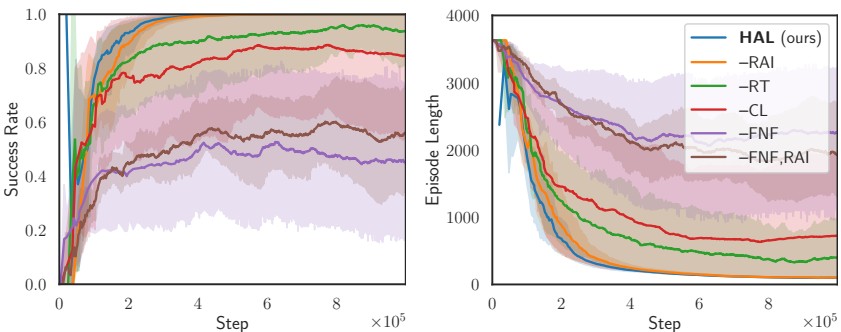

Figure 11: Training curves for component-wise ablations of HAL on TREASURE task. HAL refers to the full method, while the rest of the items displayed in the legend indicate which components are removed. −RAI no longer uses the context representation as input to the affordance classifier (the classifier is trained directly over the state). −RT no longer uses affordance classifier loss to additionally tune the context representation weights. −CL removes the contrastive loss altogether, allowing the weights to be trained solely with affordance classifier gradients. Lastly, −FNF no longer filters false negatives using the learned context representation.

## B.2 HYPERPARAMETER ROBUSTNESS

In Figure 12 we demonstrate HAL's robustness to modifications of the most significant newly introduced hyperparameters on the TREASURE task. In only two cases over the wide range of values we tested did all runs not converge. The first is when the upper confidence bound percentile is set too low, which results in more false negatives being left unfiltered. The second is when the affordance classifier threshold is set too low, which results in less pruning. We see that a wide range of more aggressive settings for both of these hyperparameters are reliable. We find that the standard deviation hyperparameter is not sensitive across the values we test in the non-edge-stochastic TREASURE environment, and that lower values (e.g. $\sigma = 2.0$), which we initially hypothesized might fare better with edge stochasticity but could lead to worse representations, are in actuality still effective.

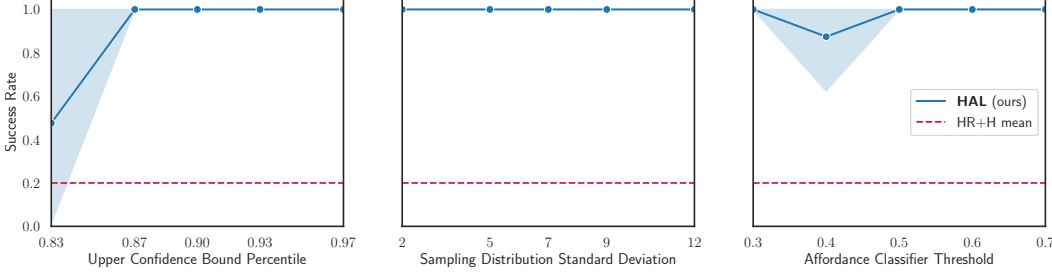

Figure 12: Plots comparing HAL's final success rate with that of HR+H's on TREASURE task with alternative hyperparameter settings. Left: percentile values used for defining the upper confidence bound $\rho$ for false negative filtering. Center: standard deviation values, $\sigma$, used for sampling positive points from the (truncated) normal distribution in the contrastive loss. Right: threshold value over which the affordance classifier output is discretized.

### B.3 VARYING σ UNDER STOCHASTICITY

In Appendix B.2, we demonstrated that in TREASURE, various values of $\sigma$ centered around the value we used in that setting ($\sigma = 7.0$) are all conducive to good performance on that task. In Figure 13, we plot the results of using different values in a stochastic version of the CRAFTING environment in the `iron` task. Although all values lead to significantly better performance than HR+H, the value we happened to used in this setting ($\sigma = 2.0$) appears to strike the best balance. Very low values (e.g. $\sigma = 1.0$) likely do not learn as general a context representation, while higher values (e.g. $\sigma = 8.0, 16.0$) are more likely to sample positive points across occurrences of edge stochasticity. It is intriguing that the final loss for $\sigma = 2.0$ is lower than for $\sigma = 1.0$. We speculate that the context learned by sampling points too close to the anchor could be a less natural representation to learn than a more general one, which considers further points.

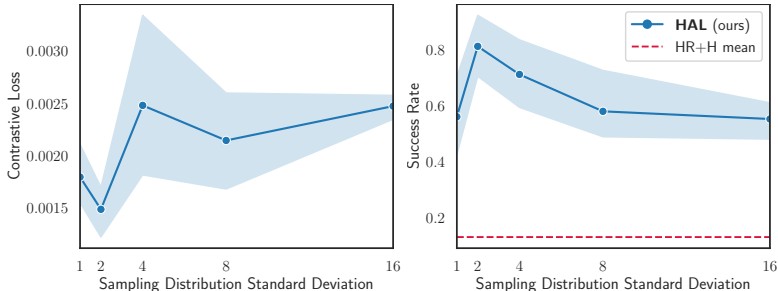

Figure 13: Plots evaluating the efficacy of various $\sigma$ values used for the contrastive loss in a stochastic CRAFTING environment on the `iron` task, with item disappearance rate $1/50$ steps. Left: final contrastive loss values for HAL. Right: final HAL success rates compared to HR+H.

## C EPISODE LENGTH PLOTS

In Figure 14 we display the episode length plots corresponding to the success rate results shown in Figures 4 and 5.

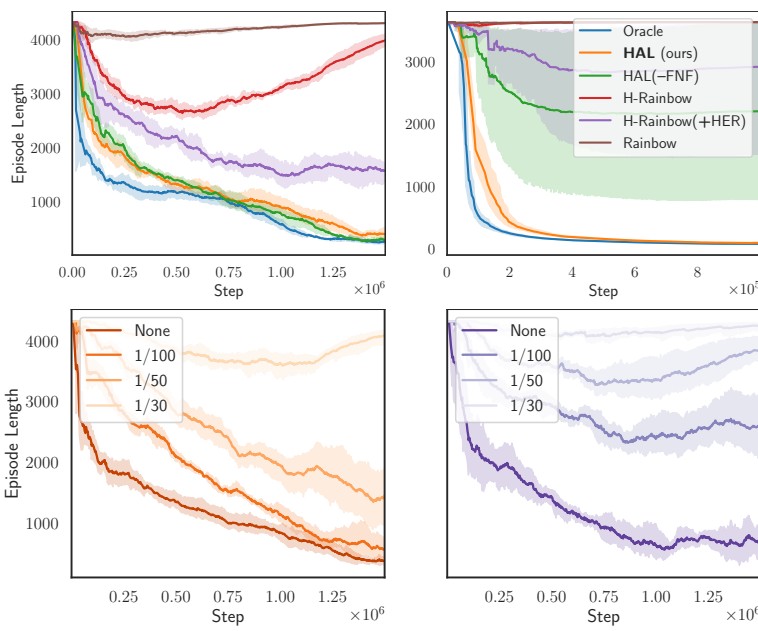

Figure 14: Episode length over the course of training for all baselines on CRAFTING `iron` (top left) and TREASURE (top right), corresponding to Figure 4. Episode length with variable levels of stochasticity CRAFTING `iron` (bottom), corresponding to Figure 5.

# D    AFFORDANCE PLOTS

We provide metrics tracked over the course of training for affordance masking as well as false negative filtering in Figure 15. The mask metrics (Figure 15a for CRAFTING `iron` and Figure 15c for TREASURE) consist of (from left to right):

- Affordance Classifier Accuracy: The accuracy of the affordance classifier w.r.t truth.
- Mask Impact: The percentage of times that the affordance mask prevents selecting an subtask that would have otherwise had the highest Q-value.
- Pruning Percentage: Percentage of subtasks pruned.
- Overpruning Percentage: Percentage of subtasks that are afforded but are pruned.
- Underpruning Percentage: Percentage of subtasks that are not afforded but are not pruned.

The false negative filtering metrics (Figure 15b for CRAFTING `iron` and Figure 15d for TREASURE) consist of:

- Filtering Margin: L2 distance at which we consider negatives to be true negatives.
- True Negative Accuracy: Percentage of true negatives falling above filtering margin.
- False Negative Accuracy: Percentage of false negatives falling below filtering margin.
- Percentage False Negatives: Percentage of negatives that are false negatives.
- False Negatives Flagged: Percentage of negatives that are flagged as false by our margin.

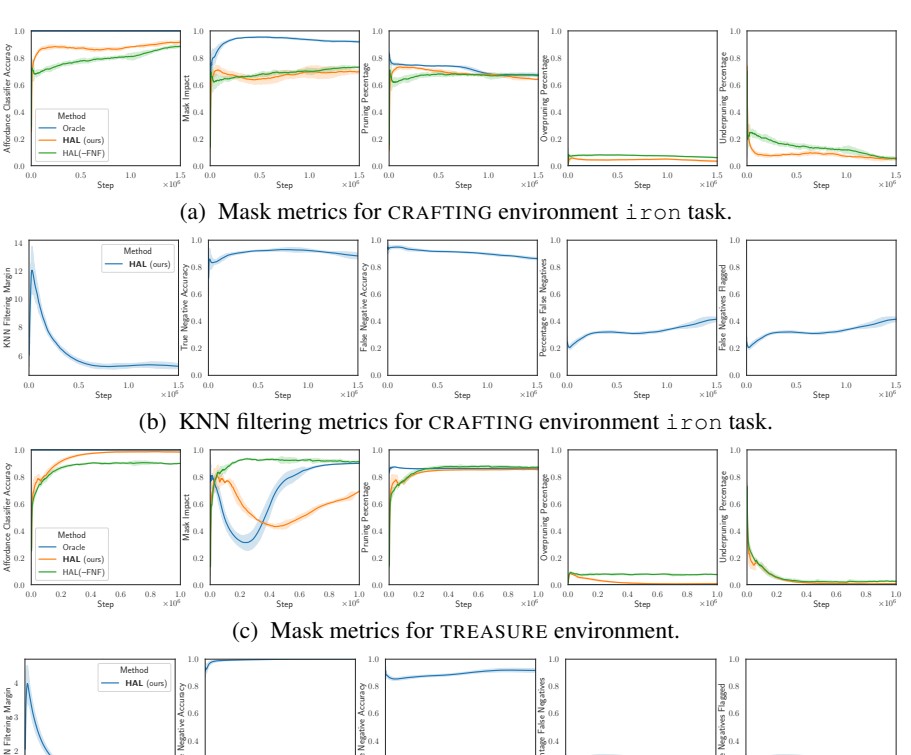

(a)  Mask metrics for CRAFTING environment `iron` task.

(b)  KNN filtering metrics for CRAFTING environment `iron` task.

(c)  Mask metrics for TREASURE environment.

(d)  KNN filtering metrics for TREASURE environment.

Figure 15

# E  ALGORITHM

---

**Algorithm 1** Learning procedure for HAL

---

1: Initialize replay buffer for meta-controller ($D_{\mathrm{mc}}$), controllers ($D_{\mathrm{c}}$), positive affordance examples ($D_g^+$), potential negative examples ($D_g^-$) for each goal $g \in \mathcal{G}$, and parameters for meta-controller ($\Theta$), controllers ($\theta$), achievement context function ($\psi$), and affordance classifier ($\phi$).
2: `steps` $\leftarrow 0$
3: **while** `steps` $<$ `max_steps` **do**
4:     `env_steps`, `option_steps` $\leftarrow 0$
5:     Receive initial state, $s$, from the environment
6:     **while** $s$ is not terminal **or** `env_steps` $<$ `max_env_steps` **do**
7:         **if** `env_steps` $= 0$ **or** `option_steps` $>$ `max_option_steps` **or** $\sum_j \mathbf{b}_j > 0$ **then**
8:             `option_steps` $\leftarrow 0$
9:             **if** `env_steps` $> 0$ **then**
10:                 Store transition $(s_{\mathtt{init}}, g, \mathbf{b}_g, s)$ in $D_{\mathrm{mc}}$
11:                 Store examples $s_t \,|\, t \in [\mathtt{init}, \mathtt{env\_steps}]$ in positive buffers $D_j^+$
12:                     where $\mathbf{b}_j = 1$ and negative buffers $D_k^-$ where $\mathbf{b}_k = 0$
13:             **end if**
14:             `init` $\leftarrow$ `env_steps`
15:             $\mathbf{f}^s \leftarrow f_\phi(z_\psi(s))$                           ▷ Get predicted affordances
16:             $\mathbf{q} \leftarrow Q_{\mathrm{mc}}(s, \cdot; \Theta)$                  ▷ Compute Q-values for all goals
17:             $\mathbf{q}_j \leftarrow -\infty \,\forall\, \mathbf{f}_j^s = 0$                 ▷ Mask non-afforded goals
18:             $g_{\max} \leftarrow \arg\max \mathbf{q}$                        ▷ Get best *afforded* goal
19:             $g_{\mathrm{aff}} \leftarrow Rand(j \,|\, \mathbf{f}_j^s = 1)$             ▷ Get random *afforded* goal
20:             $g_{\mathrm{rand}} \leftarrow Rand(\mathcal{G})$                        ▷ Get random goal
21:             $g \leftarrow g_{\mathrm{aff}}$ w.p $\epsilon_{\mathrm{aff}}$, $g_{\mathrm{rand}}$ w.p $\epsilon_{\mathrm{mc}}$, else $g_{\max}$       ▷ $\epsilon^2$-greedy goal selection
22:         **end if**
23:         $a_{\max} \leftarrow \arg\max Q_{\mathrm{c}}(s, \cdot; g, \Theta)$        ▷ Get best action – conditioned on goal
24:         $a_{\mathrm{rand}} \leftarrow Rand(\mathcal{A})$                          ▷ Get random action
25:         $a \leftarrow a_{\mathrm{rand}}$ w.p $\epsilon_{\mathrm{c}}$, else $a_{\max}$               ▷ $\epsilon$-greedy action selection
26:         `steps` $++$, `option_steps` $++$, `env_steps` $++$
27:         Send action $a$ to the environment and receive next state $s'$, rewards $r$, and milestones $\mathbf{b}$
28:         Store transition $(s, a, r, s', g)$ in $D_{\mathrm{c}}$
29:         $s \leftarrow s'$
30:         **if** `steps` $\%$ `update_freq` $= 0$ **then**
31:             Sample batch of transitions from $D_{\mathrm{c}}$ and use it to update $\theta$ via Q-learning loss
32:         **end if**
33:         **if** `steps` $\%$ `meta_update_freq` $= 0$ **then**
34:             Sample batch of transitions from $D_{\mathrm{mc}}$ and use it to update $\Theta$ via Q-learning loss
35:         **end if**
36:         **if** `steps` $\%$ `margin_update_freq` $= 0$ **then**
37:             Sample disjoint sets of positive examples from $D_g^+$ and update negative filtering
38:                 margin
39:         **end if**
40:         **if** `steps` $\%$ `aff_update_freq` $= 0$ **then**
41:             Sample positive and negative examples from $D_g^+$ and $D_g^-$ respectively to update $\phi$
42:                 with binary cross entropy loss (ignoring any negatives with computed distance
43:                 scores less than false negative filtering margin)
44:         **end if**
45:         **if** `steps` $\%$ `rep_update_freq` $= 0$ **then**
46:             Sample anchor, positive, and negative states from $D_{\mathrm{c}}$ to update $\psi$ via contrastive loss
47:         **end if**
48:     **end while**
49: **end while**
50:

---

# F IMPLEMENTATION DETAILS

All experiments are run with 5 random seeds each. We use 4 parallel environments for data collection with all methods. We do not include Noisy Nets or C51 in our implementation of Rainbow, as we do not find them to improve performance in our domains and only increase training time. Hyperparameters are shown in Table 2. All update frequency parameters are computed with respect to total environment steps (across parallel environments). Relative to the controller's Q-learning updates, all other algorithmic mechanisms are updated less frequently for the sake of efficiency, at rates which do not appear to affect the performance of our method. The meta-controller, affordance classifier, and the context representation are all trained every 10 controller updates. Since the optimal separation margin between false negatives and true negatives changes slowly over time, and each computation is expensive, each classifier head's margin is updated every 150 controller updates.

Table 2: Hyperparameters used in HAL and baselines

| Name | Description | Value |
|------|-------------|-------|
| adam_lr | Adam learning rate (across all networks) | 0.000625 |
| adam_eps | Adam epsilon | 0.00015 |
| batch_size | Training batch size | 32 |
| $\lambda$ | Discount factor | 0.99 |
| targ_update | Target network update frequency | $10^3$ |
| exp_steps | Exploration steps before updates | 400 |
| $\epsilon_c$ | Epsilon used in controller's $\epsilon$-greedy procedure | $0.5 \rightarrow 0.05$ |
| $\epsilon_{mc}$ | Epsilon used in meta-controller's $\epsilon$-greedy procedure | $0.2 \rightarrow 0.05$ |
| $\epsilon_{aff}$ | Affordance eps. used to randomly select within mask | $0.8 \rightarrow 0.00$ |
| n_steps | # of steps used for $n$-step returns | 10 |
| max_option_steps | Maximum option steps before timeout | 50 |
| update_freq | Freq. of controller weight updates | 4 |
| meta_update_freq | Freq. of meta-controller weight updates | 40 |
| margin_update_freq | Freq. that false negative filtering margins are updated | 600 |
| aff_update_freq | Freq. of affordance classifier weight updates | 40 |
| rep_update_freq | Freq. of context representation weight updates | 40 |
| $\sigma$ | Standard deviation used for inter-context sampling | 7.0 |
| $\sigma_{stoch}$ | $\sigma$ used for edge stochasticity runs | 2.0 |
| knn_n | # of points used for sampling for KNN procedure | 1000 |
| knn_k | # of nearest neighbors used for KNN procedure | 1 |
| fnf_conf | Confidence value used for upper confidence bound | 0.95 |
| fnf_perc | Percentile value used for upper confidence bound | 0.9 |

