# OpenReview forum: "Possibility Before Utility: Learning And Using Hierarchical Affordances"
_ICLR.cc/2022/Conference — ICLR 2022 Spotlight_

### Official Review · Reviewer_m66y · 2021-10-21

**Correctness:** 3
**Technical Novelty And Significance:** 3
**Empirical Novelty And Significance:** 3
**Recommendation:** 8
**Confidence:** 4

**Main Review:**

Overall an interesting paper with a simple idea, yet a fairly complicated algorithm to achieve the envisioned gains. The paper is well written and polished (although I have spotted several grammatical errors here and there) and the experimental evaluation is nice and fairly extensive and includes proper baselines. Related work is done fairly well too, and up-to-date. The conceptual steps (state abstraction which is induced from data, milestones and the f-function, their interdependence, and the mix with HRL) are clear although quite some practical choices are made and it is hard to see how robust these choices are (e.g. can the method diverge with a poor choice for some of the elements/hyperparameters?). Overall, I see several reasons for accept, but I also have a couple of things that are not yet entirely clear to me:
1) The option framework is chosen early, but after that the formalization is not appealing to the simple option formalism at all. I think it would be more clear to define the milestones etc in an option framework first, and only then go to the deep approximations. Now things are mixed, and not clear enough. A more thorough definition of the approach and the underlying (semi)MDP first would be preferred.
2) Related to the options, the affordances --as modeled here-- are actually some indication of a mix of option completion prediction and estimating the applicability of subpolicies, but not entirely clear enough (also not in the context of related work). Work like Ketharpal et al uses affordances purely for limiting action selection (ie. what is used here too) but based on " intended effects", whereas here it is about "possibility to obtain the milestone/effect" but it is also about "completion" of subpolicies and this is less clearly defined. Even more; the paper is unclear about the relation between subgoal policies and the vector induced by the f-function. The last sentence of section 6.1. says that for this domain "the set of milestones contains each object the agent can successfully interact with" whereas in the caption of Fig 4 it is said that "sub-policies, on average, receive the correct milestone when called", and earlier in the paper I wrote in the margin something like "f(.) =?= #options???" meaning that in the right part of Fig 2 f() is used as a mask over Q-values (of subpolicies) but does this mean there is a milestone for each subpolicy? This triggers the question again about "option completion prediction" (and related work) but at least it signifies that there is something unclear about the definition of f and what the milestones are in relation to the subpolicies.
3) The experiments are nice, and informative. Yet, if affordances are modeled as they are here, as ways to mask out particular "probably unsuccessful options", they seem to mainly improve sample complexity, with as cost the increased computational complexity. That is, the 4 parallel learning tasks are more complex than a standard HRL method (like H-Rainbow here). I wish more would be said about this tradeoff. In addition, I think that some kind of ablation study on the four learning tasks would be useful: they do influence each other very much, hence if one or more would be absent (or stable by using an oracle) it would be easier to see how they help and interrelate. Also, because many practical choices have been made (and thresholds, and hyperparameters been chosen) I would like to see how robust the method is over these choices (but the general case for this is maybe out of scope for this paper).
4) Based on some of the unclarity in 1-3, the comparison to related work is also not entirely clear. The comparison against recent LTL-style representations in RL is no strong anyway, since these are very different settings (with very different aims and goals). I do see that the authors try to make the case that they do not need complex symbolic structures, and there is some overlap with the specifications of (hierarchical) tasks, but the work in this paper is much more practical on a deep HRL method with an extra signal (milestones) and I do not see a deep connection. Also: the supply of the milestones can also be quite a bias and something one has to deliver as input (although, indeed, HAL can also work if this specification is incomplete). One aspect that could be made more explicit is the link between "milestones" in this paper, and "task monitors" as used by LTL-style work, in which several papers feed these into a deep RL system to enrich the state with "information about progress on a task" (which is, to me, very related). I would expect also more connection to HRL and the discovery of options and subgoals maybe.

In addition, some other issues are raised throughout the paper:
- Caption Fig 1 grammar error ("defines").
- There is no real definition of "affordance" in this paper. It is only implied, even though there are many definitions in the literature.
- Section 5, 2nd sentence is key: "restriction of subtask selection". It would be good to introduce this more strongly earlier in the paper, since many other affordance models (e.g. in robotics) are about probabilistic models (between actions, objects and effects).
- Page 5: line end-3: "to collected" (grammar)
- The part about context learning is ok (page 6) but if there is space, it could be expanded to say more about the rationale and the underlying technical components to make this work.
- Page 6: explain "KNN" and connect it to the method.
- The experiments are well done, and "learning efficacy", "robustness" (twice) are clear, but the task-agnostic setting is not at all.
- Fig 5 shows how relative the performance is/can be (see also my remarks above about the experiments). With half of the milestones, H-Rainbow+HER is as good as HAL. Turning it around: HAL needs at least half of the milestones to be better. Not much is said in the paper about this.
- Fig 5 also shows the stochasticity results, but I prefer a more thorough explanation in terms of how the algorithms work.

**Summary Of The Paper:**

This paper is in the realm of hierarchical reinforcement learning, and the aim is to employ a particular kind of affordance model: milestones of subtasks which can be achieved and a coupling between states and "affordable" subgoal completion (from that state). Basically, this makes it possible to limit action choices (or more general: bias choices) to actions/subpolicies for which it is known from experience that they are in reach from this state with this action/subpolicy. The paper introduces the HAL method, in which an indicator function for the milestones, a state generalization function (mapping similar affordance abilities to similar abstract states), and low level and hight level Q-functions are all learned in parallel. Using two domains, the method is evaluated and put against competetive baselines and it is shown that there are benefits in general, but also for specific aspects such as inexact milestone knowledge or stochasticity.

**Summary Of The Review:**

Overall, the approach is interesting, and valid, and significant. There are some unclarities about particular aspects (see review), and I think that the positioning can be done better, but my feeling is that they do not hinder acceptance too much.

---

> ### Author Response · Authors · 2021-11-18
> **Detailed response to reviewer m66y (part 1)**
>
> > The conceptual steps (state abstraction which is induced from data, milestones and the f-function, their interdependence, and the mix with HRL) are clear although quite some practical choices are made and it is hard to see how robust these choices are (e.g. can the method diverge with a poor choice for some of the elements/hyperparameters?).
>
> Please see item (6) of our general response---we demonstrate robustness to hyperparameters, and that HAL performs best when all components introduced in our work are present. While all components are necessary for best performance, most combinations still perform better over the HR+H baseline.
>
> > 1. The option framework is chosen early, but after that the formalization is not appealing to the simple option formalism at all. I think it would be more clear to define the milestones etc in an option framework first, and only then go to the deep approximations. Now things are mixed, and not clear enough. A more thorough definition of the approach and the underlying (semi)MDP first would be preferred.
>
>  We mention the options framework early to set the stage for temporally extended actions in the form of subtasks, and so that readers familiar with the framework (as we believe many will be) will be able to understand our approach in a familiar context. As early as Section 2 we discuss that our model of affordances is used to effectively alter the initiation set of options, and in the  "Method overview" paragraph in Section 5, we discuss the various termination conditions of each option. However, we decided to omit a more detailed discussion of the options framework due to space constraints, and because we did not believe it was necessary to fully understand our approach. We appreciate this feedback, but we have respectfully decided to keep mostly the same structuring.
>
> > 2. Related to the options, the affordances --as modeled here-- are actually some indication of a mix of option completion prediction and estimating the applicability of subpolicies, but not entirely clear enough (also not in the context of related work). Work like Ketharpal et al uses affordances purely for limiting action selection (ie. what is used here too) but based on " intended effects", whereas here it is about "possibility to obtain the milestone/effect" but it is also about "completion" of subpolicies and this is less clearly defined. Even more; the paper is unclear about the relation between subgoal policies and the vector induced by the f-function. The last sentence of section 6.1. says that for this domain "the set of milestones contains each object the agent can successfully interact with" whereas in the caption of Fig 4 it is said that "sub-policies, on average, receive the correct milestone when called", and earlier in the paper I wrote in the margin something like "f(.) =?= \#options???" meaning that in the right part of Fig 2 f() is used as a mask over Q-values (of subpolicies) but does this mean there is a milestone for each subpolicy? This triggers the question again about "option completion prediction" (and related work) but at least it signifies that there is something unclear about the definition of f and what the milestones are in relation to the subpolicies.
>
> For the work by Khetarpal et al., intents are defined as consequences of low level actions. For our work, the "intent" of each subtask is to achieve the corresponding milestone. We have now made this clear in the related work section while discussing affordances. We have also updated the wording to make it clearer that subtasks are represented by milestone symbols. In both environments we roughly designate each successful object interaction as a subtask (*mine this*, *craft that*, *open door*, *collect key*, etc.). We have added examples to the TREASURE environment explanation (last sentence of section 6.1) to clarify. Please also see item (5) of our general response.

---

> ### Author Response · Authors · 2021-11-18
> **Detailed response to reviewer m66y (part 2)**
>
> > 3. The experiments are nice, and informative. Yet, if affordances are modeled as they are here, as ways to mask out particular "probably unsuccessful options", they seem to mainly improve sample complexity, with as cost the increased computational complexity. That is, the 4 parallel learning tasks are more complex than a standard HRL method (like H-Rainbow here). I wish more would be said about this tradeoff. In addition, I think that some kind of ablation study on the four learning tasks would be useful: they do influence each other very much, hence if one or more would be absent (or stable by using an oracle) it would be easier to see how they help and interrelate. Also, because many practical choices have been made (and thresholds, and hyperparameters been chosen) I would like to see how robust the method is over these choices (but the general case for this is maybe out of scope for this paper).
>
> We observe that certain tasks cannot be tackled at all without the additional learning tasks (e.g. HR+H shows no sign of converging anytime soon on TREASURE), so the trade-off is clear in these cases. These tasks do slightly increase the average step time compared to HR+H, but as we now discuss in Appendix F, each learning task we introduce is able to occur at relatively infrequent intervals (compared to the controller updates). For example, the margin for each classifier head is updated every 150 controller updates. We have now included an ablation study as well as hyperparameter robustness results---please see item (6) of our general response.
>
> > 4. Based on some of the unclarity in 1-3, the comparison to related work is also not entirely clear. The comparison against recent LTL-style representations in RL is no strong anyway, since these are very different settings (with very different aims and goals). I do see that the authors try to make the case that they do not need complex symbolic structures, and there is some overlap with the specifications of (hierarchical) tasks, but the work in this paper is much more practical on a deep HRL method with an extra signal (milestones) and I do not see a deep connection. Also: the supply of the milestones can also be quite a bias and something one has to deliver as input (although, indeed, HAL can also work if this specification is incomplete). One aspect that could be made more explicit is the link between "milestones" in this paper, and "task monitors" as used by LTL-style work, in which several papers feed these into a deep RL system to enrich the state with "information about progress on a task" (which is, to me, very related). I would expect also more connection to HRL and the discovery of options and subgoals maybe.
>
> We agree that many recent LTL-style approaches in RL have different goals compared to our setting (e.g. specifying reward functions, safety), but we also believe there is a deep connection between many existing HRL approaches and LTL-style (automata) approaches, even if the connection is not made explicit. Approaches like that use policy sketches (e.g. Andreas et al., 2017) or subtask preconditions (e.g. Sohn et al., 2020) implicitly assign an automata to each task. Each task completion symbol updates the subtask dependency context deterministically. Reward machines (Icarte et al., 2020) do this explicitly. As to the final point, please see item (4) of our general response.
>
> > There is no real definition of "affordance" in this paper. It is only implied, even though there are many definitions in the literature.
>
> We appreciate this feedback, but have respectfully decided to omit any formal definition for this term. We provide a rough definition in the introduction so readers understand the general sense of the word, but in generally only discuss it insofar as it helps us get to our specific definition of "hierarchical affordances". We did not believe it was necessary to precisely define the overloaded term "affordances" itself, since it is only used as a stepping stone to our flavor of (hierarchical) affordances, which we do provide a precise definition for.
>
> > Section 5, 2nd sentence is key: "restriction of subtask selection". It would be good to introduce this more strongly earlier in the paper, since many other affordance models (e.g. in robotics) are about probabilistic models (between actions, objects and effects).
>
> In the abstract and last paragraph of the introduction we discuss the pruning of subtasks in reference to hierarchical affordances. Is there somewhere else we might be able to emphasize this to improve clarity?
>
> > Page 5: line end-3: "to collected" (grammar)
>
> Corrected!

---

> ### Author Response · Authors · 2021-11-18
> **Detailed response to reviewer m66y (part 3)**
>
> > The part about context learning is ok (page 6) but if there is space, it could be expanded to say more about the rationale and the underlying technical components to make this work.
>
> Please see item (2) of our general response. We have also added many other clarifications throughout Section 5, which should hopefully make the technical components clearer. Additionally, we are in the process of cleaning and documenting the code we linked to in our document, which should also help.
>
> > Page 6: explain "KNN" and connect it to the method.
>
> Please see item (1) of our general response (while improving the clarity of the "False negative filtering" section, we removed the term "KNN" altogether).
>
> > The experiments are well done, and "learning efficacy", "robustness" (twice) are clear, but the task-agnostic setting is not at all.
>
> The essence of this section is that no extrinsic rewards are used, so the meta-controllers are not trained. HAL is able to use knowledge of affordances to explore the high-level subtask space more effectively to reach items deeper in the hierarchy. HAL randomly selects subtasks estimated to be afforded, while HR+H randomly selects across all (since it has no notion of affordances, and cannot do better without any extrinsic supervision). We are happy to further attempt to clarify this setting of any aspects remain unclear.
>
> > Fig 5 shows how relative the performance is/can be (see also my remarks above about the experiments). With half of the milestones, H-Rainbow+HER is as good as HAL. Turning it around: HAL needs at least half of the milestones to be better. Not much is said in the paper about this.
>
> Please see item (3) of our general response---we now provide an explanation for this.
>
> > Fig 5 also shows the stochasticity results, but I prefer a more thorough explanation in terms of how the algorithms work.
>
> Please see item (2) of our general response, as well as new changes to this paragraph.

---

### Official Review · Reviewer_3eiv · 2021-10-27

**Correctness:** 3
**Technical Novelty And Significance:** 3
**Empirical Novelty And Significance:** 2
**Recommendation:** 8
**Confidence:** 3

**Main Review:**

# Strengths
- The central idea of the paper is very good.
- The paper is well written and structured.
- The motivation is clearly explained and I liked the running example with the pasta making to explain some of the concepts. As I was reading it, I was hoping to find an experiment with an agent using the approach to learn to make pasta. That would have been a nice example.
- I like Figure 1 that explains the different approaches to the problem.

# Weaknesses
- One of the advantages of the approach proposed is that you could potentially tackle problems with more complex action spaces. I'm not 100% convinced that the two example domains (Crafter and Treasure) are necessarily the best domains to show what the approach can do.
- In the introduction (page 2, 2nd paragraph) you discuss the completion of tasks. You mention that if a task is not completed some approaches need to introduce a symbol that indicates that tasks are undone. However, there are existing approaches (especially in the planning and agent communities) which can specifically deal with goals not being achieved. In fact in the BDI (beliefs, desires, intentions) agent literature there are agent programming languages that can specifically handle plan failure.
- Something that was just briefly touched on in the paper was the relationship between an affordance and an agent's intent. If we look Gibson's, Chemero's and the work of Norman in affordance theory from the cognitive sciences literature we see that the relationship between an agent's goals, their intentions and the affordances available to an agent; these are deeply linked. It would have been good to explore this a bit more especially since this a link to HRL where goals and sub-goals play a much more prominent role compared to regular RL. Perhaps something to look at for future work.
- This is just a minor point, but it is perhaps more accurate to say that your model was inspired by theory of affordances (or perhaps a model of affordances) rather than saying you have incorporated affordances into your approach.
- One of the arguments for introducing affordances into these type of algorithms (and you have made a similar argument in this paper) is that it has the potential to reduce the action space the agent needs to explore. This would imply that by incorporating affordances in a HRL model that my learning times for a particular task/game could be reduced. Comparing the learning times for the same task using the same RL algorithm (with and without affordances) would have been nice to see. This would have been a nice complement to Figure 4.
- Evaluation needs some additional work. A discussion of limitations and future directions is needed.

# Questions / Issues
- Many references are missing publication details. For example (Kulkarni 2016), the conference where this paper was published is not shown. This is the case for many of the references and should be fixed before the paper can be accepted.
- Question about the terminology "milestone". Is this basically goal or sub-goal achievement?
- First sentence in Section 5. Reads a bit off. I think you might have missed a word in this sentence. "...for selecting any subtasks the others."
- It wasn't 100% clear to me what you meant by the oracle function. I noticed this also in the results when you are plotting the Oracle against other RL algorithms. How is the oracle computing which action to take?

**Summary Of The Paper:**

This paper incorporates the concept of affordances (the idea that humans and agents perceive the world in terms of relevant action possibilities) from ecological psychology with hierarchical reinforcement learning. While affordance theory has been previously incorporated into agent action-reasoning-perception models and more recently with reinforcement learning, this work specifically looks at incorporating the concept with HRL. This is significant because many complex problems need to be broken down into subtasks. One of the challenges of many RL domains is the large action spaces that agents have to explore. While HRL helps to reduce this burden by breaking down a complex task into sub-tasks the incorporation of an affordance concept can further reduce the action space which the agent needs to consider. This paper combines these two ideas and has the potential to allow researchers to use RL in more challenging domains.

The paper develops a theoretical framework for combining affordances with HRL and conducts a number of experiments in two simple grid world domains allowing for comparison with baseline RL algorithms.

**Summary Of The Review:**

- I think this is a good paper. There are some issues that need to be addressed but not enough to not recommend it for acceptance.
- The central idea is a good one. I would say this is an incremental improvement to existing work, but has the potential to reduce the action space that agents have to explore and potentially reduce learning times.

---

> ### Author Response · Authors · 2021-11-18
> **Detailed response to reviewer 3eiv (part 1)**
>
> > One of the advantages of the approach proposed is that you could potentially tackle problems with more complex action spaces. I'm not 100% convinced that the two example domains (Crafter and Treasure) are necessarily the best domains to show what the approach can do.
>
> We agree that if a difficult task can be decomposed into subtasks, our approach would likely yield benefits in these settings. However, for more complex low-level action spaces, the limiting factor would probably be the low-level reinforcement learning algorithms used to train the subpolicies. We kept our domains simple to prevent confounding variables, but we hope that future might push the true limits of our algorithm on tasks with greater (e.g. visuomotor) complexity.
>
> > In the introduction (page 2, 2nd paragraph) you discuss the completion of tasks. You mention that if a task is not completed some approaches need to introduce a symbol that indicates that tasks are undone. However, there are existing approaches (especially in the planning and agent communities) which can specifically deal with goals not being achieved. In fact in the BDI (beliefs, desires, intentions) agent literature there are agent programming languages that can specifically handle plan failure.
>
> We would appreciate it if the reviewer could link to BDI work that may be relevant to our setting. In this paragraph we are attempting to contrast approaches that rely on subtask completion symbols in order to determine the current context, with our approach, which also utilizes these symbols to improve training, but does not fully rely on them to compute the current context. In both cases, these symbols are received after subtask completion, not before, so we are unsure if this fully corresponds to plan failure---please let us know if we are mistaken. Any method that uses symbols to indicate context will require "undo" symbols. We will try to make this discussion clearer once we are able to dive into relevant literature on BDI.
>
> > Something that was just briefly touched on in the paper was the relationship between an affordance and an agent's intent. If we look Gibson's, Chemero's and the work of Norman in affordance theory from the cognitive sciences literature we see that the relationship between an agent's goals, their intentions and the affordances available to an agent; these are deeply linked. It would have been good to explore this a bit more especially since this a link to HRL where goals and sub-goals play a much more prominent role compared to regular RL. Perhaps something to look at for future work.
>
> Fantastic point, and we agree that this link ought to be explored more in future work. We made the link more explicit by explaining in Section 2 that "in our work, high-level intents are implicitly defined by the specification of milestones".
>
> > This is just a minor point, but it is perhaps more accurate to say that your model was inspired by theory of affordances (or perhaps a model of affordances) rather than saying you have incorporated affordances into your approach.
>
> We agree with this assessment, but are unsure where specifically in the manuscript we can improve clarity on this point.
>
> > One of the arguments for introducing affordances into these type of algorithms (and you have made a similar argument in this paper) is that it has the potential to reduce the action space the agent needs to explore. This would imply that by incorporating affordances in a HRL model that my learning times for a particular task/game could be reduced. Comparing the learning times for the same task using the same RL algorithm (with and without affordances) would have been nice to see. This would have been a nice complement to Figure 4.
>
> By fixing the number of steps over which the exploration schedule is annealed for each algorithm, and seeing that HAL performs significantly better within that time-frame, we believe we have in essence demonstrated that HAL is able to perform well given less learning time. For some of our tasks, reaching nearly 100% success rate (like HAL does) would require training for an exceedingly expensive amount of time.
>
> > Evaluation needs some additional work. A discussion of limitations and future directions is needed.
>
> Please see items (3-6) in our general response---we more thoroughly discuss limitations and future work.
>
> > Many references are missing publication details. For example (Kulkarni 2016), the conference where this paper was published is not shown. This is the case for many of the references and should be fixed before the paper can be accepted.
>
> We have fixed this issue.
>
> > Question about the terminology "milestone". Is this basically goal or sub-goal achievement?
>
> Now addressed in Section 4---yes, they may be thought of as subtasks or subgoals.

---

> ### Author Response · Authors · 2021-11-18
> **Detailed response to reviewer 3eiv (part 2)**
>
> > First sentence in Section 5. Reads a bit off. I think you might have missed a word in this sentence. "...for selecting any subtasks the others."
>
> We have fixed this grammatical issue.
>
> > It wasn't 100% clear to me what you meant by the oracle function. I noticed this also in the results when you are plotting the Oracle against other RL algorithms. How is the oracle computing which action to take?
>
> We have improved the clarity in Section 6 to address this confusion. The oracle learns a meta-controller and controller the same way as HAL, but instead of using an affordance classifier to mask the meta-controller, the ground-truth affordances (computed manually from the environment) are used.

---

### Official Review · Reviewer_QUoN · 2021-10-29

**Correctness:** 4
**Technical Novelty And Significance:** 4
**Empirical Novelty And Significance:** 4
**Recommendation:** 8
**Confidence:** 4

**Main Review:**

Strengths:
- The paper addresses an important problem and contributes real progress.
- The paper is mostly really well written, using clear wording and notation.
- The results, though limited in quantity, are indicative and make a solid case for the method.

Weaknesses:

Not many, none serious.

I'm not sure I fully understand the contrastive loss function; its description is definitely too terse.  Unlike earlier, $N$ does not appear to be the segment length but some constant.  The anchor is apparently chosen from the current segment, but this is not stated.  What does it mean for $s_j^p$ to be chosen "according to a segment-truncated normal distribution"? I guess it is its rank within the segment that is chosen from a normal distribution centered at the rank of the anchor point.  This should also be reflected in the mean of the normal distribution; $a$ does not make sense.  And shouldn't $s_j^a$ be excluded from this choice?

"We show in Figure 5 that this parameter is relatively easy to set": If all runs shown used the same parameter value then the results are suggestively reassuring, but they do not show how performance would differ if (say) specially-tuned values had been used.

Some Details:

- The fact that the milestone dependency structure must be pre-specified becomes clear pretty late into the paper.  This should be noted in the introduction, if not in the abstract.
- The elements $b_g^t$ (4th row of Sec. 4) are the same as the elements $b_{g,t}$ near the bottom of p. 3 but the latter uses a lower instead of an upper index.  Reading the paper, it is quite obvious that this is in fact the same $b$, but it would be nice to make this connection explicit since it is key to understanding the method.
- Two crippled sentences on p. 5:
  - "subtask the others"
  - "failed to collected"
- p. 6, "Context learning": $f^*(s_i) = f^*(s_j)$ should read $f^*(s_t) = f^*(s_{t+1})$
- p. 6, "set used in the KNN procedure": I found this wording misleading and suggest its replacement by "$k$ closest positive points in $D_g^+$", or perhaps simply "$d_q^k$".
- Fig. 5 left: It would be interesting to know why the HAL curve drops sharply upon removal of the 5th milestone.
- Fig. 5 center and right: it would be interesting to know how policies trained without edge stochasticity would perform in environments with various levels of edge stochasticity.
- Fig. 6: Quantify the depth of each milestone within the task hierarchy.

**Summary Of The Paper:**

Given a task hierarchy specified as a dependency graph of milestones, this work employs a two-level hierarchical reinforcement-learning method similar to Kulkarni et al. 2016.  The lower level learns to achieve a given milestone in a classical fashion, while the upper level learns to choose milestones such that ultimately the agent will achieve the overall goal.  The core contribution is a method for pruning candidate milestones (upper-level actions) that are not currently achievable (afforded to the agent by the environment), using an affordance classifier trained on the fly.

**Summary Of The Review:**

The paper addresses an important problem and proposes a novel and effective solution that is likely to be used and built upon.  "Do not exist in prior works" is a strong statement; there is certainly work this paper builds upon, but, if one draws the line around the "contributions" tightly enough, I cannot point to specific prior work in which "aspects of the contributions exist".

---

> ### Author Response · Authors · 2021-11-18
> **Detailed response to reviewer QUoN**
>
> > I'm not sure I fully understand the contrastive loss function; its description is definitely too terse. Unlike earlier, $N$ does not appear to be the segment length but some constant. The anchor is apparently chosen from the current segment, but this is not stated. What does it mean for $s_j^p$ to be chosen "according to a segment-truncated normal distribution"? I guess it is its rank within the segment that is chosen from a normal distribution centered at the rank of the anchor point. This should also be reflected in the mean of the normal distribution; $a$ does not make sense. And shouldn't $s_j^a$ be excluded from this choice?
>
> We have updated the explanation surrounding the triplet loss formulation with the following changes. We have removed the unnecessary indexing on the sum (which is meant to represent the loss over a batch of samples). We removed the $a$ in the distribution notation, clarified the meaning of segment truncation, and added that we do indeed exclude the anchor point from the choice of $s_j^p$. Also, there is no notion of "current segment" when it comes to the anchor---the anchor is selected randomly across all encountered segments, and we then choose a positive and negative example accordingly.
>
> > "We show in Figure 5 that this parameter is relatively easy to set": If all runs shown used the same parameter value then the results are suggestively reassuring, but they do not show how performance would differ if (say) specially-tuned values had been used.
>
> Please see item (6) of our general response---we now include these experiments.
>
> We have addressed the minor syntactic errors referenced in the third and fourth bullet points. The rest of the detailed feedback is addressed below.
>
> > The fact that the milestone dependency structure must be pre-specified becomes clear pretty late into the paper. This should be noted in the introduction, if not in the abstract.
>
> We have updated the final introductory paragraph to make this point clearer. However, it should be noted the milestones themselves are assumed to be pre-specified, but not the dependency structure.
>
> > The elements $b_g^t$ (4th row of Sec. 4) are the same as the elements $b_{g,t}$ near the bottom of p. 3 but the latter uses a lower instead of an upper index. Reading the paper, it is quite obvious that this is in fact the same $b$, but it would be nice to make this connection explicit since it is key to understanding the method.
>
> We have now made this link explicit in Section 3. However, we still do not use $t$ as a superscript for $b_{g, t}$ to avoid notational confusion in the equation (since the superscript for $\gamma$ refers to exponentiation).
>
> > p. 6, "set used in the KNN procedure": I found this wording misleading and suggest its replacement by "k closest positive points in $D_g^+$", or perhaps simply "d_q^k".
>
> Please see item (1) of our general response---we have clarified this section.
>
> > Fig. 5 left: It would be interesting to know why the HAL curve drops sharply upon removal of the 5th milestone.
>
> Please see item (3) of our general response---we now offer an explanation for this.
>
> > Fig. 5 center and right: it would be interesting to know how policies trained without edge stochasticity would perform in environments with various levels of edge stochasticity.
>
> We agree this would be an interesting experiment! We have not run this experiment yet, but will consider implementing this setting and obtaining results before the end of the rebuttal period.
>
> > Fig. 6: Quantify the depth of each milestone within the task hierarchy.
>
> We have added a new axis and plotted the depth of each milestone in the hierarchy. See updated Figure 6, as well as discussion in item (5) of our general response.

---

> > ### Comment · Reviewer_QUoN · 2021-11-19
> > **OK.**
> >
> > Thank you, you addressed all of my points to my satisfaction. And I am looking forward to the results of the new experiment :-)

---

### Official Review · Reviewer_LfMQ · 2021-11-02

**Correctness:** 3
**Technical Novelty And Significance:** 3
**Empirical Novelty And Significance:** 3
**Recommendation:** 8
**Confidence:** 3

**Main Review:**

Strength:
1. I like the idea of learning affordance and using them to prune impossible subtasks during exploration.
2. Grounding on the current state, the affordance prediction can get rid of the history of subtask completion and handle stochasticity.
3. The paper proposes an "achievement context" trick that is used to filter out false-negative samples during affordance classification.
4. The paper presents experiments that demonstrate the effectiveness of the proposed method and provides extensive ablation studies.
5. The overall writing is easy to follow.

Weakness and comments:
1. Why affordance classifier is learning faster than the meta-controller? Do we need to collect tons of samples to train an affordance classifier? When coming to a novel state, why is the classifier able to make a correct affordance prediction? Could you provide more intuition or motivation behind the affordance classifier?
2. For methods using "atomic propositions", it seems that they can also handle the stochasticities if they model probabilistic transitions instead of deterministic transitions.
3. "each additional milestone must correspond to a behavior that is necessary for achieving the final milestone" confuses me. Why is this true? Can we have redundant milestones?
4. For CRAFTING, "The full set of milestones contains items that are either craftable or collectable." Do the milestones include the number of items? What if we need multiple blocks of wood to make a pickaxe?
5. Currently, the milestones are manually defined, which involves human prior. Do you have any idea how to select milestones automatically? This may be important for the proposed method to be generalized to various tasks.

6. page 3: $\mathcal{G}$ appears but is not explained. (explained on page 4).
7. page 3: Could you use another symbol to replace $b_g$, since there is another $b_g$ on page 4.
8. page 4: $\mathbb{E}\left[\sum_{t=0}^{N} r_{t}+\max _{g^{\prime}} Q_{m}^{*}\left(s_{N}, g^{\prime}\right) \mid s_{0}=s, g_{0}=g, a_{t} \sim \pi_{g}, s_{t} \sim P\right]$. what is $g_0$ here? Is $N$ fixed and why is it so?
9. page 4: what is $T$ in $t_{0}<t<T$?
10. page 4: Does $b_{g}^{t}$ indicate the completion of subtask at time t only, or also include the completion before t.
11. "Crucially, unlike atomic propositions, the history of milestones alone need not define which subtasks are currently possible." confuses me.
12. "we have achieved something notable without having to assign it utility." confuses me.

**Summary Of The Paper:**

The paper proposes Hierarchical Affordance Learning (HAL) that predicts sub-task affordance grounding on the current state. The affordance prediction can be used to prune impossible subtasks leading to a more efficient exploration.

The paper proposes a trick called "predicting achievement context, " which filters out false-negative samples during affordance classification.

The paper presents experiments on CRAFTING and TREASURE that demonstrate the proposed method's effectiveness and provides extensive ablation studies.

**Summary Of The Review:**

I like the idea of this paper, and the experiments also convinced me.

---

> ### Author Response · Authors · 2021-11-18
> **Detailed response to reviewer LfMQ (part 1)**
>
> > 1. Why affordance classifier is learning faster than the meta-controller? Do we need to collect tons of samples to train an affordance classifier? When coming to a novel state, why is the classifier able to make a correct affordance prediction? Could you provide more intuition or motivation behind the affordance classifier?
>
> We hypothesize in Section 6 that the affordance classifier learns faster since it is trained on a (relatively simple) supervised learning task, whereas the meta-controller requires (expensive) temporal difference updates. The affordance classifier is trained with the same data that fills the controller's replay for Q-learning. We have added a few sentences to the "Method overview" paragraph in Section 5 to address your final two questions regarding intuition (item (2) of our general response).
>
> > 2. For methods using "atomic propositions", it seems that they can also handle the stochasticities if they model probabilistic transitions instead of deterministic transitions.
>
> We now mention this possibility in Section 1, but we note that although probabilistic transitions can handle an incomplete symbolic *signal*, they still requires a full *set* of symbols to be defined, as well as predefined contexts.
>
> > 3. "each additional milestone must correspond to a behavior that is necessary for achieving the final milestone" confuses me. Why is this true? Can we have redundant milestones?
>
> Two redundant (additional) milestones would still be useful compared to no milestones, but not compared to two unique milestones. We now include "unique" in parentheses to imply that each additional behavior should be unique for the set to be *maximally* useful.
>
> > 4.  For CRAFTING, "The full set of milestones contains items that are either craftable or collectable." Do the milestones include the number of items? What if we need multiple blocks of wood to make a pickaxe?
>
> Milestones do not contain the number of items received. The agent looks at the state to determine the effect of each milestone. This allows our method to, for instance, handle cases where trees give agents variable amounts of logs, etc. More environment details are now included in Appendix A, including the numerical aspect of subtasks in CRAFTING.
>
> > 5.  Currently, the milestones are manually defined, which involves human prior. Do you have any idea how to select milestones automatically? This may be important for the proposed method to be generalized to various tasks.
>
> We agree that is an important task, but we leave it for future work. We now cite some relevant option / subgoal discovery approaches in Section 7 that might be adapted to identifying milestones automatically (item (4) of general response).
>
> > 6. page 3: $\mathcal{G}$  appears but is not explained. (explained on page 4).
>
> We now introduce this explicitly as the "goal space".
>
> > 7. page 3: Could you use another symbol to replace $b_g$, since there is another $b_g$ on page 4.
>
> They are in fact related, and we've now made the connection explicit in Section 3 paragraph 2.
>
> > 8. page 4: $\mathbb{E}\left[\sum_{t=0}^{N} r_{t}+\max {g^{\prime}} Q{m}^{*}\left(s_{N}, g^{\prime}\right) \mid s_{0}=s, g_{0}=g, a_{t} \sim \pi_{g}, s_{t} \sim P\right]$ What is $g_0$ here? Is $N$ fixed and why is it so?
>
> Options may be terminated for a number of reasons in our setting, so that length will vary. We now make it clear it is not a constant.
>
> > 9. page 4: what is $T$ in $t_0 < t < T$?
>
> We added the word "future" when introducing $b_g^T$ to indicate that $T$ is used to index some arbitrary future time point (and we do eventually say that $t_0 < t < T$).
>
> > 10. page 4: Does $b_g^t$ indicate the completion of subtask at time $t$ only, or also include the completion before $t$.
>
> We have updated the wording to "completed *on* time-step $t$" (to indicate only at time $t$) to replace the more ambiguous "at".
>
> > 11. "Crucially, unlike atomic propositions, the history of milestones alone need not define which subtasks are currently possible." confuses me.
>
> Since we assume that context can change in ways not indicated by the symbols (i.e. node and edge stochasticities which are not indicated by milestones), works relying on symbols like atomic propositions (e.g. work on reward machines by Icarte et al.) to determine the current context cannot handle this, unless some sophisticated form of probabilistic transitions are modeled. In our PASTA example, for instance, we would like to update our context (which defines which subtasks are currently possible) when we drop our noodles on the floor without requiring that we receive a symbol for this occurrence.

---

> > ### Comment · Reviewer_LfMQ · 2021-11-24
> > **further question**
> >
> > Thanks for the detailed response. I have one more question about the additional milestone. "Each additional milestone g′ added to G is useful as an intermediate signal so long as g′ corresponds to a unique behavior necessary for achieving g_K ." Can we have additional milestones that are useful for achieving an intermediate milestone g*, but g* may be different from the final goal g_K?

---

> > > ### Author Response · Authors · 2021-11-24
> > > **Response**
> > >
> > > Yes, that is correct. If an additional milestone $g$ is a unique and useful behavior for achieving $g^*$ (where $g^*$ itself is a unique and useful behavior for achieving $g_K$), we still consider $g$ (by proxy) a useful behavior for achieving the final milestone $g_K$.
> > >
> > > Another way of thinking about it is that if $g$ is a useful milestone for achieving $g^*$, and we remove $g^*$ from the milestone set, $g$ will still remain a useful milestone for achieving $g_K$; the milestone dependencies will simply have changed, as well as the required behaviors for each subpolicy (loosely speaking, the subpolicy for $g_K$ will now be responsible for doing behaviors that were previously relevant for $g^*$). For example, in PASTA, if "boil water", "cook noodles", and "mix noodles and sauce" are the milestones, removing "cook noodles" milestone means that after "boil water" is done, "mix noodles and sauce" subtask will first need to cook the noodles, and then perform the original behavior.
> > >
> > > We demonstrate this in our "Robustness to milestone selection" experiments, where HAL is able to successfully learn over of diverse sets of milestones that are strict subsets of the "complete" human-designed set. It might also be helpful to refer to Figures 7 and 8 in Appendix A, which show (abstract) milestone hierarchy diagrams of the TREASURE and CRAFTING tasks.
> > >
> > > We hope this helps, and please let us know if we can clarify anything further. Thanks!

---

> ### Author Response · Authors · 2021-11-18
> **Detailed response to reviewer LfMQ (part 2)**
>
> > 12.  "we have achieved something notable without having to assign it utility." confuses me.
>
> We tried to maintain generality here, but one can think of milestones as subtasks or subgoals---this is indeed how we use them later in the work. We have updated the section to state this specifically. Milestones increase the density of the overall signal given to the agent, but they are not directly incorporated into the (extrinsic) reward function. This is in contrast to reward shaping methods, which directly incorporate new rewards into the original reward signal. In this case, the "utility" or scaling assigned to the new rewards must be balanced with that of the original rewards. In our method, none of the milestones are assigned utility, and instead the meta-controller eventually learns this automatically.
>
> A minor point, but the current Correctness score of 1 appears to be incongruous with the sentiments expressed in the review. We kindly ask the reviewer to double-check this score.

---

### Author Response · Authors · 2021-11-18
**General response**

We thank the reviewers for their detailed and insightful feedback on our paper. We have incorporated each reviewer's suggestions to the best of our ability, and believe our updated draft is much stronger for it. All major edits are highlighted in blue. In this comment we address changes made in response to points raised by multiple reviewers. All other points are addressed in replies to the reviews themselves.

1. We have clarified some confusions that multiple reviewers mentioned in the "False negative filtering" paragraph in Section 5.
2. We have added two sentences better explaining the intuition of our method (i.e. what happens when a novel state is encountered?) in the "Method overview" paragraph in Section 5.
3. We better address the "drop" in performance once 5 milestones are removed (Figure 5). We note that this is due to the increased sparsity of the signal and variance in the milestone set quality (e.g. since smaller sets of random milestone combinations will have larger gaps on average). More generally, the fewer milestones the agent is able to use, the closer we are to the flat RL setting, which struggles on these difficult tasks, and may in general require more training time. To better demonstrate that HAL can make better use of high quality sets with 5 milestones removed, we double the training time, and find that HAL is able to attain a $97$% success rate on one of its sets, while HR+H's maximum final success rate (which happens to be on the same set) is only around $70$%.
4. We have expanded Section 7 to more thoroughly address limitations and (as in most cases, corresponding) directions for future work.
5. We have added task hierarchy diagrams for both environments in Appendix A, as well additional environment details. We have also included a second axis in Figure 6 with a corresponding line plot indicating the (approximate) hierarchy depth of each subtask, which corresponds to the labels in Figure 7.
6. Lastly, we have run component-wise ablations and hyperparameter robustness experiments, and the results are now discussed in Appendix B. In summary, all components in HAL are shown to be necessary, HAL is able to succeed using a wide range of values for the most significant hyperparameters, and while a wide range of $\sigma$ values (i.e. standard deviation for sampling positive points for contrastive learning) lead to successful learning under stochastic conditions (compared to baseline), a wise setting of this parameter, which takes into consideration the balance between sampling diverse positive pairs and avoiding sampling over edge-stochastic events, can lead to even better performance. We updated the "Context learning" paragraph in Section 5 to address these new results.

Note, we have also made a handful of small changes scattered throughout the document to address figure formatting issues, syntactic errors, and other minor issues independently identified.

---

### Comment · Reviewer_m66y · 2021-11-20
**general comments**

Thanks to the reviewers for their extensive comments. I think they are adequate and very informative. My evaluation was a choice between two "accept" options, but based on the other reviews and the extensive rebuttal, I have changed it to the lower of the two to signal one aspect. I still like the paper, and I think it could be accepted, but the answers about options, affordances and milestones point to something I picked up while reading: that the method is not well described in terms of existing formalisms (and that not much will be repaired for the final version). I think that if the paper would have been written as an proper "extension" of the option framework, with proper definitions of affordances and milestones, it would have been a much better paper, much more easy to read, and more importantly, much more clear in its contributions. Now it is up to the reader to figure this out and relate it to a large amount of related work, and this generates more confusion than necessary.

---

> ### Author Response · Authors · 2021-11-23
> **Restructuring changes**
>
> We thank reviewer *m66y* for elaborating on the concerns expressed in their original review. After further consideration, we have decided to follow through with a restructuring of the kind they have suggested. Most significantly, we have now described the options framework in detail and have formally situated both milestones and hierarchical affordances within this framework. Only after this foundation is laid do we then discuss the deep training procedures. We believe not only that the core contributions of our paper are now more clear, but that many of the minor confusions raised by other reviewers are also more adequately addressed. The updated draft is attached to this submission with all major additions highlighted in **green**. Below is a summary of these changes.
>
> - A description of the *options framework* and the resulting SMDP is now described in detail in Section 3 ("Preliminaries").
> - In the first sentences of Section 4 we make the connection between *milestones*, *subtasks*, *options*, and *subpolicies* clearer.
> - The middle third of Section 4, Paragraph 1 now explains the precise role that milestones play in our method, and their connection to (1) the options framework (for us, as subgoals to train the options using the *milestone signal*) and (2) the *intents* used to define *affordances* the work of Kulkarni et al. (for us, as *high-level* intents to define *hierarchical affordances*).
> - The final third of Section 4, Paragraph 1 describes the requirements of the milestone set more clearly, referencing prior work on options.
> - The second half of Section 4, Paragraph 2 formally connects the subtask pruning concept to the restriction of the corresponding option's initiation set. Importantly, we provide the high-level intuition that, by restricting the option initiation set by the definition of hierarchical affordances, we in effect impose a subtask dependency structure on top of the options framework.
> - We made the content now in Section 4, Paragraph 3 more concise to accommodate the aforementioned additions, which we believe also has improved the clarity of this content.
> - In Section 4 we also distinguish the model of affordances (modeling successful milestone attainment, using the milestone signal) from models of option completion used in previous methods, which model all option completion dynamics (not only success). We explicitly describe the full set of termination conditions when the algorithm as a whole is described in Section 6, as they are relevant only to the training procedure and not the hierarchical affordances formulation.
> - Previously, the controllers were introduced before milestones, and created some confusion that multiple reviewers voiced. That section (Section 5) now discusses milestones in the context of controllers (which follows Section 4, on "Milestones And Hierarchical Affordances").

---

> > ### Comment · Reviewer_m66y · 2021-11-28
> > **an interesting turn of events**
> >
> > I was (hopefully happily) surprised by these changes. They sound like it would/could counter the criticism I used to justify me lowering my grade a little still. I still need to have a look at the paper to evaluate the changes, but it sounds promising. Will have a look soon.

---

> > > ### Comment · Reviewer_m66y · 2021-11-29
> > > **Final comment**
> > >
> > > I had to go trough the paper again to look at the above extensions, and to see again how things connect. I like the new additions concerning options, since they make it much easier to see how fairly simple the notions used are. I think it is now more clear relative to the existing literature and formalisms. What is still missing, now even more maybe, is a better positioning against HRL methods that learn the hierarchy, identify bottlenecks, or somehow learn particular aspects while learning the task. I hope some things will still be added/modified in the related work section. My evaluation is still roughly a 7/10 but reviewers are forced to either choose below or above that, and in the context of all reviews and the efforts the authors have put in the new version, I will raise my evaluation again to my original.

---

### Decision · Program_Chairs · 2022-01-20

**Decision:**

Accept (Spotlight)

**Comment:**

This paper proposes a hierarchical reinforcement learning approach that exploits affordances to better explore/prune the subtasks, and thus making the overall learning more efficient.

The idea of the paper is novel and interesting.

After the rebuttal, all the reviewers agree that the paper is a solid contribution.
Therefore, I recommend acceptance of this paper.